# Self-Replicating RNA Viruses for Vaccine Development against Infectious Diseases and Cancer

**DOI:** 10.3390/vaccines9101187

**Published:** 2021-10-15

**Authors:** Kenneth Lundstrom

**Affiliations:** PanTherapeutics, 1095 Lutry, Switzerland; lundstromkenneth@gmail.com

**Keywords:** self-replicating RNA viruses, vaccines, infectious diseases, cancer, immune response, tumor regression, protection, approval

## Abstract

Alphaviruses, flaviviruses, measles viruses and rhabdoviruses are enveloped single-stranded RNA viruses, which have been engineered for recombinant protein expression and vaccine development. Due to the presence of RNA-dependent RNA polymerase activity, subgenomic RNA can replicate close to 10^6^ copies per cell for translation in the cytoplasm providing extreme transgene expression levels, which is why they are named self-replicating RNA viruses. Expression of surface proteins of pathogens causing infectious disease and tumor antigens provide the basis for vaccine development against infectious diseases and cancer. Self-replicating RNA viral vectors can be administered as replicon RNA at significantly lower doses than conventional mRNA, recombinant particles, or DNA plasmids. Self-replicating RNA viral vectors have been applied for vaccine development against influenza virus, HIV, hepatitis B virus, human papilloma virus, Ebola virus, etc., showing robust immune response and protection in animal models. Recently, paramyxovirus and rhabdovirus vector-based SARS-CoV-2 vaccines as well as RNA vaccines based on self-amplifying alphaviruses have been evaluated in clinical settings. Vaccines against various cancers such as brain, breast, lung, ovarian, prostate cancer and melanoma have also been developed. Clinical trials have shown good safety and target-specific immune responses. Ervebo, the VSV-based vaccine against Ebola virus disease has been approved for human use.

## 1. Introduction

Vaccine development has always had a central position in prevention of infectious diseases, but with the onset of the COVID-19 pandemic it has reached unprecedented levels. Similarly, the area of cancer vaccines has drawn plenty of attention. Obviously, the development of vaccines against SARS-CoV-2 has been approached from every possible angle including inactivated and attenuated viruses, protein and peptide subunit-based vaccines, nucleic acid-based vaccines, and viral vectors [1]. In this review the focus will be on viral vectors. Although the strongest progress has been achieved for adenovirus vectors with Emergency Use Authorization (EUA) for the ChAdOx1 nCoV-19 [2], Ad26.COV2.S [3], and rAd26-S/rAd5-S [4], only vaccine candidates based on self-replicating RNA viruses will be described here. In addition to SARS-CoV-2, other viral pathogens such as influenza virus, human immunodeficiency virus (HIV), hepatitis B virus (HBV), human papilloma virus (HPV), Ebola virus (EBOV) and Lassa virus (LASV) have been targeted [5]. Self-replicating RNA viruses have also been used for cancer vaccine development. In this review multiple examples of immunization with self-amplifying RNA viral vectors expressing various antigens against infectious agents and tumors are presented. The advantages and disadvantages of using self-replicating RNA viral vectors, especially RNA-based delivery, are also discussed.

## 2. Self-Replicating RNA Viruses

Application of self-replicating RNA viruses for vaccines against infectious diseases and cancer has clear advantages compared to other viral vectors and non-viral delivery systems. Self-replicating RNA viruses deposit their RNA directly into the cytoplasm of infected host cells [6], which requires no delivery to the nucleus as is the case for some other RNA viruses such as influenza virus, and also for DNA-based delivery. In the case of positive strand RNA viruses such as alphaviruses, the most significant feature relates to the efficient self-replication/amplification of delivered RNA by the established RNA replication complex, which can accumulate close to 10^6^ copies of subgenomic RNA per cell in the host cell cytoplasm [7]. It will generate high levels of antigen expression, which can potentially elicit superior immune responses and may also allow immunizations with smaller doses resulting in reduced adverse events. It can also provide extreme expression of toxic, anti-tumor and immunostimulatory genes for cancer vaccination and therapy. Additionally, self-replicating RNA viral vectors can be utilized as recombinant replication-deficient viral particles, replicon RNA, or layered DNA/RNA vectors (Figure 1). Another feature of interest is the transient nature of high levels of transgene expression provided by self-replicating RNA viruses due to the degradation of RNA transcribed from recombinant particles and RNA replicons within 5–7 days post-immunization. It is advantageous for vaccine development against both infectious diseases and cancers. Furthermore, in contrast to for instance retroviruses, alphavirus RNA is not subjected to reverse transcription and integration into the host genome.

Self-replicating RNA viruses can be divided into two groups based on the polarity of their RNA genome. All self-replicating RNA viruses possess a single-stranded non-fragmented RNA (ssRNA) genome. However, alphaviruses [6] and flaviviruses [8] have a positive-sense RNA genome, whereas the genome of paramyxoviruses [9] and rhabdoviruses [10] is of negative polarity. The difference in polarity has consequences for their applications as the positive sense ssRNA is immediately after infection translated in the cytoplasm. In the case of alphaviruses, expression systems are based on delivery of recombinant viral particles, RNA replicons or plasmid DNA replicons. Infection with recombinant particles and electroporation or lipid-based transfection of in vitro transcribed replicon RNA deliver positive sense ssRNA to the cytoplasm of host cells. Utilization of plasmid DNA transfection requires initial delivery of DNA to the nucleus followed by in vivo transcription of RNA. The recombinant RNA containing the non-structural replicase genes and the gene of interest (GoI) is efficiently amplified (self-replication) from a minus strand RNA template and translation of recombinant protein coding for the GoI occurs in the cytoplasm. A schematic illustration of alphavirus self-replicating expression systems is presented in Figure 1. The most prominent alphavirus expression systems are based on Semliki Forest virus (SFV) [11], Sindbis virus (SIN) [12] and Venezuelan equine encephalitis virus (VEEV) [13]. Flavivirus expression systems have been engineered for Kunjin virus (KUN), where the gene of interest is introduced between the first 60 nucleotides of the C20 core protein and the last 22 codons of the E22 envelope protein [14]. The GoI is expressed as part of a larger polyprotein from which the flanking regions are cleaved off by the FMDV-2A protease sequence in the KUN vector [15]. KUN production has been facilitated by the engineering of a packaging cell line [16]. In addition to KUN, expression vectors have been engineered for West Nile virus (WNV) [17], yellow fever virus (YFV) [18], Dengue virus (DENV) [19], and tick-borne encephalitis virus (TBEV) [20]. Furthermore, the bovine viral diarrhea virus (BVDV) has been engineered as an expression vector by introducing the GFP reporter gene between the N(pro) and C genes of the non-cytopathic type-1 BVDV strain SD1 [21]. Similarly, expression of GFP from a bicistronic classical swine fever virus (CSFV) in infected host cells confirmed the potential of CSFV as an expression vector [22].

In the case of RNA viruses with negative ssRNA polarity such as vesicular stomatitis virus (VSV), the RNA-dependent RNA polymerase (RdRp) responsible for self-replication is encoded in the L gene and the phosphoprotein (P) is an essential cofactor for the RdRp activity. In the case of VSV expression systems, the VSV glycoprotein (G) gene is generally replaced by the GoI or the GoI is inserted between the G and L genes for the generation of either pseudotype or recombinant VSV particles (Figure 2A) [23,24]. Pseudotype VSV can be produced in mammalian cells by transfection of plasmid DNA containing foreign envelope genes followed by infection with the VSV G-complemented G-VSVΔG pseudotype virus. The generated pseudotype VSV can infect target cells, but do not produce infectious viral progeny. In contrast, infection of mammalian producer cells with VSV G-complemented recombinant virus replacing the VSV G with a foreign envelope generates fully infectious viral progeny [24]. Originally, application of reverse genetics for expression vector engineering was based on recombinant vaccinia virus vectors. In the case of VSV, the nucleoprotein (N), phosphoprotein (P), polymerase (L) and the full-length antigenomic RNA were expressed from four plasmids under the control of the T7 promoter from a vaccinia virus [25]. To develop a vaccinia virus-free system, the VSV N, P and L genes were introduced downstream of both the T7 promoter and an internal ribosomal entry site (IRES) and the T7 polymerase was provided by the BSR-T7/5 stable cell line. The elimination of the vaccinia virus from the reverse genetics system presented an attractive alternative for generation of infectious VSV from DNA.

In the case of rabies virus (RABV), the GoI can be inserted between the N and P genes and G and L (Figure 2B) [26,27]. A vaccinia virus-free reverse genetics system has also been engineered for RABV [28]. For paramyxoviruses, measles virus (MV) vectors, packaging systems and helper cell lines have been engineered to allow rescue of replicating MV from plasmid DNA vectors [29,30]. Generally, the GoI is inserted between the phosphoprotein (P) and the matrix protein (M) or alternatively between the hemagglutinin (HA) gene and the large protein (L) (Figure 2C). As a comparison of positive and negative strand RNA viruses, their replication strategies are illustrated in Figure 3.

## 3. Infectious Diseases

The classic approach for targeting of infectious diseases for vaccine development has comprised overexpression of immunogenic surface epitopes or proteins as antigens, which elicit immune responses leading to protection against challenges with lethal doses of pathogens [31]. A large number of preclinical and some clinical studies have been conducted for vaccine candidates based on self-replicating RNA viral vectors and it is only possible to provide some examples below and in Table 1 and Table 2. The most common targets have been viral infections, but additionally vaccine candidates against bacterial infections and tropical diseases have been developed.

Among alphaviruses, Chikungunya virus (CHIKV) [93,94] and VEEV [95] have been responsible for epidemics in Africa, Polynesia, and South America. Vaccine development has included expression of the CHIKV envelope polyprotein E3-E2-6K-E1 from a chimeric VSV vector, which elicited neutralizing antibody responses and provided protection against CHIKV in mice after a single administration of 1 × 10^7^ pfu of VSV particles [32]. In another approach, expression vectors for VEEV, western equine encephalitis virus (WEEV), and eastern equine encephalitis (EEEV) were engineered by removing the furin cleavage site at the junction between the E2 and E3 envelope proteins [33]. It prevents the cleavage of the precursor p62 into E2 and E3 to produce infectious particles but generates replication-deficient recombinant virus particles [33]. The combination of 1 × 10^7^ IU of VEEV, WEEV, and EEEV or individual viral recombinant particles induced strong neutralizing antibody responses and protected mice from subcutaneous or aerosol challenges with VEEV, WEEV, and EEEV [33]. Similarly, immunization of cynomolgus macaques with 2 × 10^8^ IU of the VEEV-WEEV-EEEV combination elicited strong immune responses and protected against challenges with VEEV and EEEV. In contrast, the immune response against WEEV was weak and the protection against challenges with WEEV was only partial [33]. In the context of DNA-based delivery, the attenuated VEEV V4020 strain was administered to BALB/c mice as a DNA/RNA layered replicon vector, which elicited robust neutralizing antibodies and protected mice from challenges with wildtype VEEV [34]. Protection against aerosol challenges with wildtype VEEV was also demonstrated in vaccinated cynomolgus macaques [35]. Furthermore, an MV-based vector expressing CHIKV capsid and envelope proteins showed strong immunogenicity and protection from viremia in macaques [36]. The MV-CHIKV VLP vaccine candidate was evaluated for safety and efficacy in a randomized, double-blind phase I clinical trial showing a seroconversion rate of 44–92% after a single dose, which reached 100% after a second immunization [96]. It was followed by a phase II study, which elicited strong neutralizing antibodies without causing any serious adverse events making it a promising CHIKV vaccine candidate [97].

Arenaviruses including such pathogens as LASV have also been targeted for vaccine development. In this context, VSV-based expression of the LASV glycoprotein complex (GPC) provided protection against LASV strains from Liberia, Mali, and Nigeria in guinea pigs and macaques immunized with 1 × 10^6^ and 6 × 10^7^ pfu, respectively [37]. MV-based GPC expression has also demonstrated protection in macaques after a single immunization with 6 × 10^6^ pfu of MV-GPC particles [38]. A randomized, placebo-controlled, dose-finding phase I trial is in progress in healthy volunteers receiving two doses of MV-LASV [98]. In another approach, the LASV GPC gene was introduced into the YFV vector between the envelope (E) and non-structural protein 1 (NS1) [39]. Immunization of guinea pigs was 80% protective, but due to instability of the full-length GPC, GP1 and GP2 subunit constructs were engineered in individual YFV vectors [40]. Combined immunization with YFV-LASV GP1 and -GP2 showed 83% protection in guinea pigs with no stability issues. However, prime-boost vaccination of marmosets failed to provide protection confirming previous findings that robust immune responses and protection seen in rodents is not necessarily reproducible in non-human primates [41]. Expression of either LASV GPC or nucleoprotein (NP) from VEEV replicons protected guinea pigs from challenges with the LASV Josiah strain [42]. However, protection was only established after three immunizations with recombinant VEEV particles. Furthermore, a multivalent VEEV vaccine encoding GPC from the distantly related LP and Josiah strains showed protection in inbred CBA/J mice [43]. VEE vectors have also been used for targeting other arenaviruses such as Junin virus (JUNV) and Machupo virus (MACV) [44]. VEEV-based expression of JUNV- and MACV-GPC, respectively, elicited humoral immune responses and provided protection in guinea pigs after immunization with 1 × 10^7^ pfu.
vaccines-09-01187-t002_Table 2Table 2Examples of clinical studies on self-replicating RNA viral vector vaccines against infectious diseases.Virus/DiseaseAntigenVectorFindingsRef.**Alphaviruses**



CHIKV/CHIK C, Env MV-CHIKV VLPs Phase I: 100% seroconversion after two doses [96] CHIKV/CHIKC, EnvMV-CHIKV VLPsPhase II: good safety, strong immunogenicity[97]**Filoviruses**



EBOV/EVD GP (Zaire strain) VSV-ZEBOV Phase III: high vaccine efficacy, protection [99,100] EBOV/EVD GP (Zaire strain) VSV-ZEBOV Phase III: high vaccine efficacy [100] EBOV/EVDGP (Zaire strain)VSV-ZEBOVErvebo approval by the FDA, EMA[101]**Flaviviruses**



ZIKV/ZVD E MV-ZIKA-E Phase I: study completed; no results available [102] ZIKV/ZVDEMV-ZIKA-RSP-EPhase I: study in progress[103]**Lentiviruses**



HIV/AIDSHIV GagVEEV-GagPhase I: trials halted, stability & documentation[104]**Coronaviruses**



SARS-CoV-2/COV S MV (TMV-083) Phase I: weak immunogenicity, trial discontinued [75,76] SARS-CoV-2/COV S VSV (V590) Phase I: weak immunogenicity, trial discontinued [78,79] SARS-CoV-2/COV S VSVΔG-S Phase I/II: study in progress [81,82] SARS-CoV-2/COVSLNP-VEEV-S RNAPhase I/II: study in progress[85]C, capsid, CHIKV, Chikungunya virus; COV, COVID-19; EBOV, Ebola virus; E & Env, envelope proteins; EVD, Ebola virus disease; GP; Glycoprotein; S, spike protein; SARS, severe acute respiratory syndrome; VEEV, Venezuelan equine encephalitis virus; VLPs, virus-like particles; VSV, vesicular stomatitis virus; ZIKV, Zika virus; ZVD, Zika virus disease.


Among filoviruses, especially EBOV has been a common target for vaccine development due to its transmissibility and the severity of Ebola virus disease (EVD) during outbreaks in 2014–2016 [105]. In one approach, the EBOV glycoprotein (GP) D637L mutant, which displays superior cleavability and shedding compared to wildtype GP, was expressed from a KUN vector [45]. Two subcutaneous doses of 1 × 10^9^ KUN-GP/D637L VLPs was sufficient to provide protection in three out of four immunized nonhuman primates. In another study, when macaques vaccinated with 5 × 10^7^ VSV-EBOV GP particles were challenged with the EBOV-Makona [46] and Zaire (ZEBOV) [47] strains, they were resistant to EVD. The VSV-based EBOV-GP vaccine (VSV-ZEBOV) has been subjected to phase III clinical evaluation in 7651 individuals with suspected EVD, of which 4123 were vaccinated immediately and 3528 received a delayed vaccination [99]. No EVD cases were detected in individuals immediately vaccinated while 16 EVD cases were observed among those receiving delayed vaccination, which indicated that the vaccination was successful. Another phase III trial confirmed the efficacy, where no new EVD cases were detected in neither the 2119 individuals who received immediate vaccination nor in the 2041 subjected to a 21-day delay in vaccine administration [100]. The VSV-ZEBOV vaccine has been approved by both the FDA and the EMA [101]. In the context of other filoviruses, VEEV-based expression of the Marburg virus (MARV) GP provided protection in macaques against MARV challenges after immunization with 1 × 10^7^ pfu of VEEV-MARV-GP particles [100]. A VEEV vector expressing the Sudan virus (SUDV) GP was subjected to intramuscular administration of 1 × 10^10^ particles in cynomolgus macaques, which protected the vaccinated animals against SUDV challenges [48]. Moreover, macaques immunized with VEE-SUDV GP particles provided partial protection against EBOV challenges. Macaques co-administrated with VEEV-SUDV-GP and VEEV-EBOV-GP were protected against both SUDV and EBOV challenges.

In the context of flaviviruses, BALB/c mice immunized with a single dose of VEEV particles expressing the ectodomain of the DENV envelope resulted in protective immunity against challenges with DENV [49]. Moreover, immunization of mice with 2 × 10^6^ pfu of an MV vector expressing the DENV envelope protein domain III (ED3) elicited robust immune responses and resulted in partial protection against DENV [50]. A live-attenuated chimeric YFV-DENV tetravalent vaccine (CYD-TDV) has been engineered and tested in rodent and primate models, showing no toxicity, good safety, and robust immune responses [51,52]. The CYD-TDV vaccine has been tested in endemic populations [53].

In another approach, VEEV replicon RNA was engineered to express the codon optimized Zika virus (ZIKV) membrane-envelope protein (prME) [54]. The delivery of the VEEV-ZIKV-prME replicon RNA was facilitated by formulation of nanocarrier lipids (NLCs). A single intramuscular dose of 10 ng RNA provided complete protection against ZIKV challenge. Moreover, a single immunization of mice with 1 × 10^7^ pfu of the chimeric VSV vector expressing the ZIKA-prME resulted in robust antibody responses and rendered mice resistant to ZIKV challenges [32]. A live MV-based vaccine expressing the ZIKV E gene (MV-E2) provided protection against the nonlethal ZIKV Asian strain PRVABC59 and the lethal African strain MR766 in mice [55]. Despite 100% survival complete viral clearance in the brain and reproductive tract did not occur. However, co-administration of an MV vector expressing the ZIKV NS1 gene (MV-NS1 [2]) led to complete clearance of ZIKV from the female reproductive tract and fetal protection was achieved.

Although most clinical trials conducted on ZIKV vaccines are based on inactivated or attenuated viruses and nucleic acids [106], a dose-finding phase I study with MV-ZIKA (MV-E2) was carried out in 48 individuals, however, no results have been published yet [102]. Furthermore, another phase I trial to validate the safety and immunogenicity of MV-ZIKA is in progress in healthy 18–55-year-old volunteers in Austria [103].

Although HBV vaccines have been approved [107] there is still need for new vaccine development due to the discovery of breakthrough infections, for instance. In attempts to target HBV, MV vectors have been used for the expression of the surface antigen (HBsAg) [56]. Immunization with HBV-HBsAg resulted in 50% protection of rhesus monkeys. In another approach, the SFV RNA replicon expressing the HBV middle surface envelope glycoprotein (MHB) and the core antigen (HBcAg) were packaged into a VSV G envelope [57]. Mice immunized with 1 × 10^7^ pfu of SFV-G-MHB were protected against HBV challenges, while SFV-HBcAg immunizations did not provide protection. So far clinical trials on HBV vaccines have focused mainly on DNA, live virus, and peptide-based approaches [108]. Only one adenovirus-based phase I trial has been described [109]. No published data is available for clinical applications of self-replicating RNA vector-based HBV vaccines.

The HIV/AIDS epidemic has obviously accelerated the development of vaccines against HIV. Live attenuated MV vectors expressing HIV-1 Gag like particles with a gp160 Env protein envelope elicited robust cellular and humoral immune responses in mice [58]. Much attention has been dedicated to alphavirus-based HIV vaccine development. For example, mice immunized with SFV-HIV-Env particles showed superior antibody titers compared to plasmid DNA and recombinant Env protein [59]. Furthermore, intramuscular administration of SFV-HIV-Env replicon RNA induced Env-specific immune responses in four out of five mice [60]. In another approach, immunization of mice with SFV particles expressing the Indian HIV-1C Env-Gag-Pol-RT genes elicited significant T-cell responses with higher antibody titers compared to replicon RNA immunization [61]. SFV DNA replicon delivery of HIV Env and a Gag-Pol-Nef fusion protein generated strong immune responses in immunized BALB/c mice [62]. In attempts to improve stability and delivery of VEE-HIV-1 gp140 RNA replicons, cationic nanoemulsion (CNE) formulations consisting of squalene, 1,2-dioleoyl-3-tri-methylammonium-propane (DOTAP) and sorbitan trioleate were developed [63]. In a comparative study, intramuscular injection of 50 μg of VEEV RNA-CNE elicited stronger immune responses in rhesus macaques than what was obtained for VEEV particles or MF59 adjuvanted HIV gp140 protein [110]. In the case of clinical evaluations for self-replicating RNA virus-based HIV vaccines, the safety and immunogenicity of an alphavirus replicon HIV-1 Gag vaccine (AVX101) was subjected to a double-blind, randomized, placebo-controlled trial in healthy adults [104]. The study was conducted in the US and South Africa, but it was halted due to vaccine stability issues. Another phase I trial was initiated, but it was prematurely terminated because of documentation issues encountered by the contract manufacturer. However, the study results indicated that in contrast to preclinical findings, only low levels of immune responses were elicited in humans. Measurement of anti-vector antibodies showed only modest local reactogenicity.

The importance of vaccine development against influenza virus relates to the occurrence of seasonal global outbreaks. In the context of MV, a recombinant MV AIK-C vaccine expressing the hemagglutinin (HA) protein from the influenza A/Sapporo/107/2013 (H1N1pdm) strain elicited strong immune responses in cotton rats and provided protection against challenges with influenza virus [64]. In the case of VSV, the VSVΔG vector lacking the VSV G protein was engineered to express the HA protein of the highly pathogenic avian influenza virus (HPAIV) A/Vietnam/1203/04 (VN1203) strain and the neuraminidase (NA protein) of the mouse-adapted H1N1 influenza virus A/Puerto Rico/8/34 (PR8) [65]. A single immunization of mice with VSVΔG-H5N1 provided protection against lethal H5N1 infection. In another study, a VSV-based H5N1 influenza virus vector containing the full-length hemagglutinin (HAfl) was administered as a single dose or a prime-boost regimen in mice, generating protection against lethal challenges with various H5 clade 2 viruses [66]. In the context of alphaviruses, a single dose of 1 × 10^7^ pfu of VEE-HA resulted in protection against influenza A virus isolate A/HK/156/97 challenges in chickens [67]. In another study, 10 μg of SFV-HA replicon RNA provided protection in 90% of vaccinated BALB/c mice [68]. The superiority of self-replicating replicon RNA was confirmed by demonstrating that only 1.25 μg was required to provide protection in mice compared to 80 μg needed for synthetic mRNA [69]. In a novel approach, the external domain of the influenza virus M2 protein (M2e) was introduced into the E2 membrane protein in a SIN vector, resulting in SIN particles (E2S1-M2e) with M2e expressed on its surface [70]. Mice intranasally immunized with SIN E2S1-M2e were protected from challenges with a virulent influenza A virus strain. As CSFV targets monocytes and dendritic cells (DCs) the nucleoprotein (NP) and HA genes of influenza virus were inserted into the CSFV replicon RNA (RepRNA) vector [71]. Packaging of a Rep-HA/Rep-NP mix in viral replicon particles (VRPs) was compared with polyethylenimine (PEI)-based RNA complexes and naked RepRNA in pigs. Both VRPs and PEI-RepRNA complexes elicited strong HA and NP specific humoral and cellular immune responses, whereas naked RNA induced only low-level immunogenicity. Overall, CSFV VRPs showed superior immunogenicity in pigs.

The current COVID-19 pandemic has promoted vaccine development to a new level. The breath and intensity of global activities related to vaccines have been unprecedented leading to EUA of both nucleic acid- [111,112] and viral vector-based [2,3,4] COVID-19 vaccines in approximately a year from the onset of the outbreak. As the authorized viral vector based COVID vaccines are based on adenoviruses they are not discussed here, and the focus of the current review will be on self-replicating RNA viruses. Prior to COVID-19 vaccines, both SARS-CoV and Middle East respiratory syndrome-coronavirus (MERS-CoV) have been targeted. For example, mice immunized with a VEEV vector expressing the SARS-CoV Spike (S) protein resulted in protection against SARS-CoV challenges [72]. In the context of MERS-CoV, the VSV G protein was replaced by the MERS-CoV S protein [73]. A single intramuscular or intranasal immunization with VSVΔG-MERS-CoV S elicited neutralizing antibodies and T cell responses in rhesus macaques.

Obviously due to the COVID-19 pandemic, SARS-CoV-2 has received major attention as a vaccine target. MV-based expression of the SARS-CoV-2 S protein elicited robust Th1-biased antibody and T cell responses in mice [74]. The MV-SARS-CoV-2 S vaccine candidate TMV-083 was subjected to a randomized, placebo-controlled phase I clinical trial, which based on disappointing weak immune responses in vaccinated volunteers was discontinued [75,76]. VSV vectors have also been applied for overexpression of the SARS-CoV-2 S protein [77]. Immunization of mice with VSV-SARS-CoV-2 S particles elicited neutralizing antibody responses and protected against SARS-CoV-2 related pathogenesis. In the context of clinical trials, the VSV-SARS-CoV-2 S vaccine candidate V590 was evaluated in a phase I clinical trial in 252 volunteers [78]. The immunization proved safe and showed good tolerability, but the immune responses were weaker than seen in COVID-19 patients, which justified the termination of the trial [79]. In another approach the replication competent VSVΔG vector was engineered by replacing the VSV G protein with the SARS-CoV-2 S protein [80]. A single immunization with 5 × 10^6^ pfu of VSVΔG-SARS-CoV-2 S elicited potent neutralizing antibodies and protected Syrian golden hamsters against challenges with SARS-CoV-2. In the case of clinical trials, the VSVΔG-SARS-CoV-2 S vaccine candidate is subjected to a phase I/II clinical study, where volunteers will receive a single dose of 5 × 10^5^, 5 × 10^6^, or 5 × 10^7^ pfu of VSVΔG-SARS-CoV-2 in the first part of the study [81]. As the trial is still in progress the interim experience has caught attention as other vaccines have received EUA and the ethical question has been raised whether individuals in the placebo group should be entitled to other vaccines during the study period [82]. The decision was to balance the individual risk with the common good of participants without compromising the quality of randomized studies. In the context of animal coronaviruses, the chimeric flavivirus BVDV vector was applied for the expression of a spike antigen of the porcine epidemic diarrhea virus (PEDV) [83]. Intramuscular injection of BALB/c mice elicited BVDV- and PEDV-specific antibodies and neutralized both BVDV and PEDV.

Self-amplifying RNA viruses have also been used in COVID-19 vaccine development as liposome nanoparticle (LNP) encapsulated RNA replicons. In this context, VEEV-based RNA replicons were engineered to express the prefusion-stabilized SARS-CoV-2 S RNA [84]. Intramuscular administration of LNP SARS-CoV-2 S RNA in BALB/c mice elicited robust and dose-dependent SARS-CoV-2 specific antibody responses and SARS-CoV-2 neutralization. The antibody titers were superior to those seen in recovered COVID-19 patients. The LNP SARS-CoV-2 S RNA vaccine candidate has been subjected to a randomized, placebo-controlled, dose-finding phase I/II study in healthy volunteers [85]. No results from the study are available yet. In another study, the VEEV-based self-replicating RNA (STARR)-based vaccine (LUNAR-COV19) expressing the full-length SARS-CoV-2 S protein has been evaluated in BALB/c mice [86]. A single immunization elicited strong neutralizing antibody responses and both 2 μg and 10 μg doses protected humanized ACE2 transgenic mice from mortality and measurable infection after challenges with wildtype SARS-CoV-2. Recently, SIN particles expressing the SARS-CoV-2 S protein were combined with the OX40 immunostimulatory antibody (αOX40) for intraperitoneal immunization of C57BL/6J mice [87]. A prime-boost vaccination strategy with 14 days between the two doses elicited long-lasting neutralizing antibodies and robust T-cell responses and sera from vaccinated mice inhibited the function of the SARS-CoV-2 S protein. Furthermore, immunized mice were protected against challenges with SARS-CoV-2.

In addition to viral targets, bacterial and parasite infections have also been studied. For example, SIN vectors have been applied for the expression of the *Bacillus anthracis* protective antigen (PA) [88]. Immunization of Swiss Webster mice elicited PA-specific IgG and neutralizing antibody responses and provided some protection against a lethal bacterial strain. SFV particles expressing the *Brucella abortus* translation initiation factor 3 (IF3) elicited IF3-specific IgM antibodies and T cell proliferative responses and protected immunized BALB/c mice [89].

In the context of parasites and malaria vaccines, SFV particles expressing the *Plasmodium falciparum* Pf332 antigen were subjected to immunization studies in BALB/c mice [90]. A single immunization elicited Th1-type immune responses, which were further enhanced by a second immunization. In another approach, the *Plasmodium yoelii* circumsporozoite (CS) protein class I major histocompatibility complex-restricted-9-mer epitope SYVpSAEQI was expressed from a SIN vector [91]. Subcutaneous immunization of BALB/c mice with SIN-Mal particles induced robust epitope-specific CD8^+^ T cell responses and provided protection against malaria. Recently, the *Phlebotomus papatasi* SP15-*Leishmania major* stress inducible protein 1 (PpSP15-LmSTI1) fusion protein was compared for expression in BHK-21 cells after transfection of SFV replicon RNA, SFV DNA replicons and a conventional DNA plasmid [92]. The relative expression was significantly higher for the SFV replicon RNA than both SFV and conventional DNA vectors, making it an attractive alternative for vaccine development against leishmaniasis.

## 4. Cancer

Self-replicating RNA viruses have also been applied for cancer immunotherapy and cancer vaccine development [5]. The general approach has been to introduce a tumor antigen into the self-replicating RNA viral vector for immunization studies, which have demonstrated both prophylactic and therapeutic efficacy in preclinical animal models. Other approaches have involved expression of immunostimulatory genes such as cytokines and even reporter genes. Obviously, the application of reporter genes such as GFP and luciferase allows efficient monitoring of expression. The therapeutic effect seen after administration of alphavirus vectors expressing reporter genes relates to the induction of apoptosis, but the efficacy has been inferior compared to treatment with alphavirus vectors expressing cytokines like interleukin-12 (IL-12) [113]. While cancer vaccines aim at providing protection against tumor development, oncolytic viruses possess therapeutic activity, also named virotherapy, for the treatment of existing tumors [114]. Oncolytic viruses are characterized by efficient replication in and killing of tumor cells without causing harm to normal cells, which make them attractive for cancer therapy. There are different types of engineered oncolytic viruses such as herpes simplex virus, adenovirus, vaccinia virus and reovirus. Moreover, naturally oncolytic viruses have been identified for Newcastle disease virus [115]. Among self-replicating RNA viruses, attenuated MV strains [116], engineered VSV vectors [117], and the naturally oncolytic M1 alphavirus [118] exist. Although the focus is on prophylactic and therapeutic cancer vaccines, examples of virotherapy are also included here. So far, a limited number of clinical trials have also been conducted. Examples of preclinical studies and clinical trials are presented below and summarized in Table 3 and Table 4.

Brain tumors have been the target of several studies due to the severity of disease and very few alternative options of successful treatment. For instance, SFV particles expressing endostatin were compared to a retrovirus vector expressing endostatin, and SFV-LacZ in a B16 brain tumor mouse model [119]. Intratumoral administration of SFV-Endostatin showed significant tumor growth inhibition and reduction of intratumoral vascularization, which was superior to both retrovirus and SFV-LacZ treatments. In addition, intravenous SFV-Endostatin administration generated 3-fold increase in serum levels of endostatin. In another approach, intratumoral injection of DCs transduced with SFV particles expressing interleukin-18 (IL-18) in combination with recombinant IL-12 protein therapy, elicited Th1-type immune responses and provided anti-tumor immunity in a B16 brain tumor mouse model [120]. Furthermore, a SIN DNA replicon encoding the human gp100 and mouse IL-18 was evaluated in B16-gp100 implanted brain tumor models in mice [121]. Immunization with SIN-gp100-IL-18 DNA resulted in therapeutic effects, enhanced protection of malignant brain tumors, and significantly prolonged survival rates. A chimeric VSV vector, where the VSV G protein was replaced by the CHIKV E3-E2-6K-E1 polyprotein (VSVΔG-CHIKV), showed selective infection and elimination of brain tumor cells [122]. Moreover, tumor-bearing mice showed extended survival for more than 100 days. VSVΔG-CHIKV also showed targeting in intracranial melanoma xenografts derived from patients with only minor detectable infection of normal cells. Moreover, oncolytic MV vectors expressing GFP, carcinoembryonic antigen (CEA) and sodium iodide symporter (NIS) have been shown to replicate and providing cytotoxic effects in glioma stem cells (GSCs) [123]. When GSCs transduced with MV-GFP were implanted into the right caudate nucleus of nude mice a significant prolongation of survival was obtained. Application of oncolytic replication competent SFV VA-EGFP particles was evaluated in BALB/c mice in a subcutaneous orthotopic tumor model stably expressing firefly luciferase [124]. A single intravenous administration of SFV VA-EGFP completely inhibited intracranial firefly bioluminescence and provided long-term survival in 16 out of 17 mice. The treatment was well tolerated causing no damage to heart, liver, spleen, or brain.

Applications of alphavirus vectors for cancer immunotherapy and gene therapy of brain tumors have raised some concerns due to their neurotropic nature [175]. In one approach, distribution of recombinant SFV particles (recSFV) and RNA replicons (recRNA) expressing firefly luciferase was compared in tumor-free and 4T1 mammary tumor-bearing mice [176]. Intravenous administration of recRNA resulted in primary brain targeting in both tumor-free and tumor-bearing mice. However, local intratumoral injection led to high levels of luciferase expression in tumors. Interestingly, predominant tumor targeting of recSFV was observed after low intravenous or intraperitoneal viral doses, whereas higher doses led to a broader luciferase distribution. In another approach, neuron-specific micro-RNA miRT124 sequences were introduced into the replication-competent SFV4 vector, which modified its tropism [125]. A single intraperitoneal administration of SFV4-miRT124 to C57BL/6 mice with implanted CT-2A orthotopic gliomas demonstrated significant tumor growth inhibition and provided prolonged survival.

Related to breast cancer, immunization of BALB/c mice with adenovirus particles and SIN DNA replicons expressing the HER2/neu gene inhibited A2L2 tumor growth [126]. However, if the tumor challenges took place prior to immunization, no inhibition was observed. A strategy of prime immunization with SIN DNA followed by a boost with adenovirus particles significantly prolonged the survival of mice. In another study, intradermal administration of BALB/c mice with SIN-HER2/neu DNA replicons generated robust antibody responses and required 80% less replicon DNA than conventional plasmid DNA to achieve tumor protection [127]. In another study, a novel VEEV vector expressing the extracellular domain (ECD) and transmembrane (TM) domains of HER2 (VRP-HER2) showed robust immunogenicity, both preventive and therapeutic efficacy. and control of tumor growth in a HER2 transgenic mouse model [128]. Moreover, VRP-HER2 showed good tolerance in a phase I trial in stage IV HER2 overexpressing breast cancer patients and generated partial response (PR) in one patient and continued stable disease (SD) in two other patients [170]. Additionally, a phase II trial on VRP-HER2 and pembrolizumab in 39 HER2-positive breast cancer patients is in progress [170]. In another study, 2 × 10^8^ SFV-IL12 particles and 2 × 10^7^ units of an aroC^-^
*Salmonella typhimurium* strain (LVR01) were administered to mice with 4T1 tumor nodules, which provided complete inhibition of lethal lung metastases and long-term survival in 90% of immunized mice [129]. Compared to administration of either SFV-IL12 or LVR01 alone, the synergistic effect of combination therapy presents a promising alternative for prevention and eradication of metastases in advanced breast cancer. In the case of triple-negative breast cancer (TNBC), the most aggressive breast cancer molecular subtype, Doxorubicin was demonstrated to increase the oncolytic effect of the oncolytic M1 alphavirus by 100-fold, specifically in TNBC cells in vitro and significantly inhibited tumor growth in vivo [118]. In the context of MV, reverse genetics was applied to engineer a recombinant MV named rMV-SLAMblind, which is selectively unable to use the signalling lymphocyte activation molecule (SLAM) [130]. Unlike the MV-Edmonston vaccine strain, rMV-SLAMblind used the polio virus receptor-related 4 (PVRL4) as a receptor to infect breast cancer cells showing superior oncolytic activity. In vivo studies of rMV-SLAMblind in monkeys showed no clinical symptoms, suggesting that the vector could be a promising oncolytic candidate for breast cancer therapy.

Although the recombinant protein-based human papilloma virus (HPV) vaccine Gardasil was approved by the FDA in 2006 against cervical cancer [177], there is a continuous development in this area. Recombinant MV expressing the HPV-16 L1 capsid protein was subjected to immunization studies in transgenic mice, which resulted in strong humoral immune responses [131]. In another study, the MV-HPV16 L1 capsid vaccine was compared to recombinant HPV16L1 and 18L1 protein vaccines produced in *Pichia pastoris* in immunized non-human primates [132]. Both MV- and *P. pastoris*-based vaccines induced immune responses. Prime-boosting combination immunization elicited HPV-specific total IgG and neutralizing antibodies, which was not affected by pre-existing antibodies against MV. Moreover, recombinant VSV vectors have been utilized for the expression of the cottontail rabbit papillomavirus (CRPV) E1, E2, E6, and E7 proteins and immunization of rabbits [133]. VSV-E1, E2, E6, and E7 immunizations significantly reduced papilloma volumes, the VSV-E7 being the most efficient reducing the papilloma volumes by 96.9%, which ultimately eradicated the disease. In another approach, mice bearing TC-1 syngeneic tumors were immunized with VSV-HPV E7 [134]. A single intramuscular injection of C57BL/6 mice with 5 × 10^6^ pfu of VSV-HPV E7 elicited HPV16 E7 specific T cells and displayed anti-tumor activity resulting in a 10-fold reduction in tumor volume and regression of pre-existing tumors.

Among alphaviruses, VEEV vectors have been utilized for the expression of the HPV-16 E7 protein [135]. Immunization of C57BL/6 mice elicited CD8^+^ T cell responses and protected mice from tumor challenges. In another study, an SFV vector containing the translation enhancer signal from the SFV capsid gene was engineered to express the HPV E6-E7 fusion [136]. Tumor regression and complete eradication of established tumors were observed in immunized C57BL/6 mice. The SFVenh-HPV E6/E7 vaccine candidate Vvax001 has been subjected to a phase I clinical trial in 12 individuals with a history of cervical intraepithelial neoplasia [171]. Patients received three immunizations of 5 × 10^5^, 5 × 10^6^, 5 × 10^7^, or 2.5 × 10^8^ infectious SFVenh-HPV E6/E7 particles at a three-week interval. The vaccination showed high safety and tolerability in patients with HPV-induced cancers. HPV-specific immune responses were detected in all 12 patients. SFV DNA replicons have also been employed for HPV vaccine development [137]. Intradermal immunization of mice with SFV-HPV E6-E7 DNA replicons accompanied by electroporation eradicated 85% of tumors [135]. The efficacy of DNA replicon immunization compared to conventional plasmid DNA demonstrated that a 200-fold lower dose of only 0.05 μg of SFVDNA was sufficient for therapeutic efficacy.

Colon cancers have also been targeted by self-replicating RNA viral vectors. For instance, a noncytopathic KUN vector was engineered to express the granulocyte macrophage-colony stimulating factor (GM-CSF) [138]. Intratumoral immunization of BALB/c mice implanted with CT26 colon xenografts induced CD8^+^ T cell responses, resulted in tumor regression, and in cure of 50% of vaccinated mice. SFV particles expressing the vascular endothelial growth factor receptor-2 (VEGFR-2) inhibited tumor growth, reduced tumor angiogenesis, and prevented metastatic spread in immunized BALB/c mice [139]. Additionally, combination therapy of SFV-VEGFR2 and SFV-IL-4 elicited stronger VEGFR-2 antibody responses and provided prolonged survival of vaccinated mice. Immunization with RNA replicons has also been successful, the classic example being the immunization of mice with SFV-LacZ RNA, which elicited antigen-specific CD8^+^ T cell responses [140]. A single immunization with 0.1 μg SFV-LacZ RNA provided protection against tumor challenges and therapeutic immunization prolonged survival of mice with pre-existing tumors. In a phase I clinical trial, patients with stage IV colorectal cancer received VEEV particles expressing the CEA every three weeks for four immunizations [172]. Later the study was expanded to include stage III patients. Antigen-specific effector T cells were elicited, and long-term survivors were identified suggesting prolonged overall survival. In the case of oncolytic MV vectors the expression of GM-CSF resulted in therapeutic efficacy and adaptive immune responses in a colon adenocarcinoma MC38cea model [141]. Intratumoral administration of MV-GM-CSF delayed tumor progression and prolonged survival time. Complete tumor remission was observed in one third of immunized mice and tumor re-engraftment was rejected.

Another area of opportunity is lung cancer. Human H358a non-small cell lung cancer (NSCLC) cells transduced by SFV-EGFP particles were efficiently killed and the growth of H358a spheroids was inhibited [142]. Moreover, nu/nu mice with H358a xenografts were injected with SFV-EGFP particles, which resulted in complete tumor regression in three out of seven mice. In comparison to a conditionally replicating adenovirus vector (Ad5-Delta24TK-GFP), the replication-competent SFV (VA7)-EGFP particles were locally administered to nude mice with A549 lung adenocarcinoma xenografts [143]. Mice immunized with SFV-EGFP showed superior survival compared to adenovirus-based vaccination. Systemic administration, however, did not elicit significant immune responses with either vector. In another approach, SIN-LacZ particles were intravenously administered to mice with implanted CT26.CL25 colon tumors [144]. SIN-LacZ particles induced complete tumor remission and provided long-term survival. MV vectors have also been subjected to lung cancer treatment. In this context, the oncolytic MV Hu-191 strain effectively suppressed tumor growth and significantly prolonged the survival of C57BL/6 mice implanted with Lewis lung carcinoma (LLC) cells [145]. It was demonstrated in another study that the live-attenuated oncolytic MV Schwarz strain prevented uncontrollable growth of established lung and colorectal adenocarcinomas in nude mice with xenografts [146]. Similarly, the expression of CEA from the MV Edmonston strain resulted in potent killing of lung cancer cell lines and tumor regression in mice [147]. Additionally, a VSV vector expressing interferon-β (VSV-IFNβ) reduced tumor growth in intratumorally immunized mice with H2009 and A549 lung tumors [148]. Superior efficacy was achieved by intratumoral administration of VSV-IFNβ leading to tumor regression, prolonged survival, and cure in 30% of immunized syngeneic LM2 lung tumors [148].

Melanoma is an indication that has been frequently visited for vaccine development applying viral vectors [178]. For example, immunization of C57BL/6 mice carrying B16-OVA melanoma tumors with KUN-GM-CSF particles resulted in substantial tumor regression and cure of more than 50% of treated animals [138]. Moreover, expression of the chicken ovalbumin (OVA) cytotoxic T cell lymphocyte (CTL) epitope SIINFEKL from an YFV vector induced SIINFEKL-specific CD8^+^ lymphocytes and provided protection against challenges with B16-OVA or B16F0 melanoma cells in immunized C57BL/6 mice [149]. Alphaviruses have also been used for melanoma therapy. For instance, humoral immune responses, anti-tumor activity, and prolonged survival were obtained in a B16 mouse melanoma model after immunization with VEEV particles expressing the tyrosine-related protein-2 (TRP-2) [150]. In another approach, VEEV-TRP-2 particles were combined with antagonist anti-CTL antigen-4 (CTLA-4) or agonist anti-glucocorticoid-induced tumor necrosis factor receptor (GITR) monoclonal antibodies (mAbs) [151]. Immunization with VEEV-TRP-2 and anti-GITR mAbs induced complete tumor regression in 90% of mice, whereas VEEV-TRP-2 and anti-CTLA-4 treatment resulted in tumor shrinkage in 50% of animals. In the context of DNA-based delivery, SFV DNA replicons expressing VEGFR-2 and IL-12 from one plasmid vector and survivin and β-hCG antigens from another plasmid were co-administered to mice with implanted B16 melanoma tumors [152]. The combination immunization provided superior tumor growth inhibition and prolonged survival compared to administration of either SFV DNA replicon alone. In the case of MV vectors, the oncolytic potential of the MV Leningrad-16 (L-16) strain was demonstrated to provide efficient killing of tumor cells and inhibition of tumor growth in the mel Z mouse melanoma model [153]. Related to VSV, a pseudotyped vector, where the VSV-G protein was replaced by the non-neurotropic lymphocytic choriomeningitis virus (LCMV) glycoprotein (GP), showed efficacy in subcutaneous A375 xenograft and B16-OVA syngeneic mouse tumor models, and also reduced the size of lung metastasis after systemic treatment [154].

In the context of ovarian cancer, the pseudotyped VSV-LCMV-GP vector showed oncolytic activity against A2780, HOC7, SKOV6 and other ovarian cancer cell lines and in vivo in the A2780 ovarian mouse tumor model [155]. Tumor regression was further enhanced by combination treatment with the JAK1/2 inhibitor ruxolitinib. MV vectors have also been evaluated for ovarian cancer therapy. Tumor-specific targeting has been achieved by engineering of the MV-αFR vector with a single-chain antibody (scFV) sequence for the alpha-folate receptor (αFR) [156]. Intratumoral injection of MV-αFR into mice with ovarian SKOV3ip.1 xenografts demonstrated reduced tumor volumes and prolonged overall survival. Moreover, MV-CEA and MV-NIS have been applied alone or in combination for immunization of mice implanted with SKOV3ip.1 xenografts [157]. The dual therapy was superior to either MV-CEA or MV-NIS treatment alone. The MV-CEA vector has been evaluated in a phase I clinical trial in patients with taxol and platinum-refractory recurrent ovarian cancer (RROC) [173]. The study demonstrated that intraperitoneal administration of MV-CEA was well tolerated and provided dose-dependent biological activity in heavily pre-treated patients, of which SD was achieved in 14 out of 21 patients. Alphavirus vectors have also been evaluated for ovarian cancer therapy. Combination therapy of SIN-IL-12 particles and the CPT-11 topoisomerase inhibitor irinotecan provided long-term survival in SCID mice with grafted highly aggressive ES2 human ovarian tumors [158]. In another study, C57BL/6 mice with murine ovarian surface epithelial carcinoma (MOSEC) received a prime immunization of SFV-OVA followed by boost vaccination with vaccinia virus expressing OVA (VV-OVA), which elicited OVA-specific CD8^+^ T cell immune responses and enhanced anti-tumor activity [159].

Due to the poor prognosis of pancreatic cancer patients plenty of efforts have been dedicated to the development of vaccines. The oncolytic potential of VSV vectors has been verified in highly aggressive pancreatic ductal adenocarcinoma (PDAC) [160]. In comparison to Sendai virus and respiratory syncytial virus (RSV), VSV showed superior oncolytic activity although PDAC cells were shown to be highly heterogenous to VSV susceptibility reducing the therapeutic efficacy. In another study, wildtype VSV, VSV-GFP and the oncolytic VSV-ΔM51-GFP were tested in five PDAC cell lines with (+MUC1) or without (MUC1 null) MUC1 expression [161], showing oncolytic activity independent of MUC1 expression. The VSV-ΔM51-GFP vector generated significant reduction in tumor growth in mice with implanted PDAC xenografts. The anti-tumor activity was improved when gemcitabine was co-administered with VSV. Related to MV vectors, SCID mice with KLM1 and Capan-2 pancreatic tumor xenografts were immunized with MV-SLAMBlind, which resulted in significant suppression of tumor growth [162]. In the case of alphaviruses, a phase I clinical study in pancreatic cancer patients was conducted with VEEV-CEA particles efficiently infecting DCs [174]. Repeated intramuscular injection of VEEV-CEA induced clinically relevant T cell and antibody responses, which mediated cellular cytotoxicity against tumor cells and prolonged overall survival in patients.

In the context of prostate cancer, a significant delay in tumor growth and prolonged survival was seen in a prostate PC-3 mouse model after intratumoral immunization with MV-CEA [163]. In another application, co-administration of oncolytic MV and mumps virus (MuV) vectors generated superior anti-tumor activity and prolonged survival in the PC-3 prostate cancer model compared to individual administration of MV or MuV [164]. In the context of VSV vectors, the VSV-ΔM51-GFP showed efficient replication in human DU145, and PC-3 cell lines, which induced apoptosis and killing of tumor cells [165]. In vivo, malignant cells were eradicated while normal tissue was relatively unaffected in nude mice immunized with VSV-ΔM51-GFP. The survival of immunized mice was also significantly prolonged. In another study, the oncolytic VSV-LCMV-GP efficiently infected 6 different prostate cancer cell lines [166]. Intratumoral and intravenous immunization generated long-term remission of subcutaneous tumors and bone metastases in the DU145 and 22Rv1 prostate tumor mouse models. In the case of alphaviruses, a VEEV vector expressing the prostate-specific membrane antigen (PSMA) elicited strong PSMA-specific immune responses in immunized BALB/c and C57BL/6 mice [167]. Immunization studies with VEEV expressing the six-transmembrane epithelial antigen of the protein (STEAP) has been evaluated in prophylactic and therapeutic mouse models [168]. The study demonstrated CD8^+^ T cell responses against a newly defined mouse STEAP epitope, which prolonged the overall survival of mice. Moreover, TRAMP mice immunized with VEEV particles expressing the prostate stem cell antigen (PSCA) provided long-term survival in 90% of mice at 12 months post-vaccination [169]. In the context of clinical applications, a phase I study was conducted in patients with castration resistant metastatic prostate cancer (CRPC) by immunization with either 0.9 × 10^7^ or 3.6 × 10^7^ IU of VEEV-PSMA particles [179]. Although the vaccination was well tolerated the PSMA-specific immune response was weak. To address this issue, thorough dose optimization and vector engineering should be considered.

## 5. Conclusions

In summary, numerous examples of applications of self-replicating RNA viral vectors have been presented for targeting both infectious diseases (Table 1 and Table 2) and various cancers (Table 3 and Table 4). In many cases, target-specific humoral and cellular immune responses have been obtained. In the context of cancer therapy and cancer vaccinations, inhibition of tumor growth, tumor regression and even tumor eradication have been observed. Moreover, immunized animals including mice, guinea pigs and non-human primates were protected against challenges with lethal doses of infectious agents and tumor cells. One attractive characteristic of self-amplifying RNA viruses, especially alphaviruses, is the flexibility of applying them as recombinant viral particles, RNA replicons or layered DNA/RNA vectors (Figure 1). The main feature of RNA replication/amplification has allowed similar immune responses and challenge protection to be achieved for self-replicating RNA viruses with significantly lower doses compared to conventional viral particles, synthetic RNA, or plasmid DNA. Alternatively, higher doses could potentially induce stronger immune responses. Additionally, the prolonged release of antigens expressed from self-replicating RNA contributes to B cell stimulation and immune stimulation is also enhanced by generation of double-stranded RNA intermediates in transfected cells [69]. Moreover, the rapid RNA degradation renders the heterologous gene expression transient, which is an advantage for vaccine development and cancer therapy, where high-level short-term expression is preferable. On the other hand, although not the topic of this review, self-replicating RNA virus vectors are not suitable for the treatment of chronic diseases, where long-term gene expression is required. Self-replicating RNA viruses do not possess reverse transcriptase activity and therefore do not integrate into the host genome. However, application of self-replicating RNA viral vectors also presents some disadvantages. In the case of replicon RNA, the ssRNA structure is sensitive to degradation, which demands careful handling and has required RNA encapsulation in LNPs for improved stability and delivery [84,85]. RNA-based vaccines have also stricter demands on storage and transportation temperatures. In the case of recombinant self-replicating RNA virus particles, safety concerns have been raised, requiring engineering of helper vectors for conditionally infectious particles [180] and split helper systems [181]. The use of replication-proficient and oncolytic viruses for cancer therapy also needs special attention to ensure that no damage is caused to normal tissue. For instance, alphavirus vectors showing strong neurotropism, engineering of neuron-specific miRT124 sequences restricted replication to tumor cells only, allowing efficient treatment of CT-2A gliomas in mice [125].

Self-amplifying RNA viruses have been applied for some clinical trials. So far, the numbers are significantly lower than what have been seen for adenoviruses, AAV, retroviruses and lentiviruses. However, the positive results obtained so far has encouraged further engineering of improved vectors and delivery systems and optimization of dosing and prime-boost strategies. Reflecting on the success and failure of vaccine development based on self-replicating RNA viral vectors, it is difficult to point out any vector system showing superiority over other systems. Clearly, the choice of target plays a role, especially for vaccines targeting infectious diseases. It might also be good to underline the differences between cancer vaccines and vaccines against infectious pathogens. In the case of cancer vaccines, the approach is to provide both prophylactic and therapeutic efficacy and it therefore includes in a broader meaning cancer immunotherapy. For that reason, the repertoire of expressible genes of interest is much larger and in addition to tumor antigens, anti-tumor genes, cytotoxic genes and immunostimulatory genes can be overexpressed to provide preventive or therapeutic effects. Another issue relates to the differences in cancer development and infectious diseases. While viral and bacterial outbreaks can quickly develop into epidemics and even pandemics as familiarly experienced with COVID-19, although many cancers have a high mortality rate, there is no risk of causing epidemics. For this reason, the urgency for cancer vaccines seems to be less prominent compared to infectious diseases, particularly when they have reached pandemic levels.

Related to the efficacy of vaccine development, various self-replicating RNA viral vectors have elicited high neutralizing antibody titers in immunized rodents and non-human primates. Moreover, protection has been achieved in rodents and primates against challenges with lethal doses of infectious pathogens. Similarly, immunization of rodents with cancer vaccine candidates has elicited strong immune responses and in certain cases inhibition of tumor growth and/or tumor regression have been observed. Moreover, immunized animals were protected against challenges with tumor cells. Administration of oncolytic viruses has also resulted in tumor regression and in some favorable situations total tumor eradication and cure of treated mice. Typically, transfer from animal studies, particularly studies in mice, has often struggled to generate the same efficacy in clinical trials. This phenomenon has been attributed to different delivery demands in larger animals and humans and suboptimal dosing. For this reason, we should not be discouraged by these setbacks, but continue the engineering of more efficacious delivery vectors and continue dose optimization studies. It might also be advantageous to consider canine tumor models for the following reasons. First, the bigger size of dogs compared to rodents might provide a more similar situation to establish delivery in humans. Second, naturally occurring tumors in canine models resemble more closely human cancers than induced tumors in rodent models. Third, prophylactic and therapeutic evaluation in dogs might lead to veterinary applications in support of human use although it should not be seen as a shortcut to applications in humans. In the context of MV-based vaccines, potential pre-existing immunity against MV, which cannot be assessed in studies in rodents, can compromise the efficacy of vaccines in humans.

The approval by the FDA and the EMA of Ervebo, the VSV-based vaccine against EVD, has given a glimpse of hope for this strategy and further strengthens the position of self-replicating RNA viral vectors as attractive vehicles for vaccine development. One might ask why it took so long to obtain the approval for the first vaccine based on self-replicating RNA viral vectors? There is obviously no simple answer, and although breakthrough in vector and technology development since the 1980s has been remarkable, especially the combination of omics initiatives including bioinformatics, genomics, proteomics, immunomics, and vaccinomics, all pieces had to come together to guarantee the highest possible safety level of therapeutics and vaccines.

Although efficient vaccines based on adenovirus vectors against COVID-19 have received EUA, issues related to vaccine-induced immune thrombotic thrombocytopenia (VITT) and the circulation of novel more transmissible SARS-CoV-2 variants demonstrate the need for re-engineering of novel vaccines to which self-replicating RNA viral vectors may provide a substantial contribution.

## Figures and Tables

**Figure 1 vaccines-09-01187-f001:**
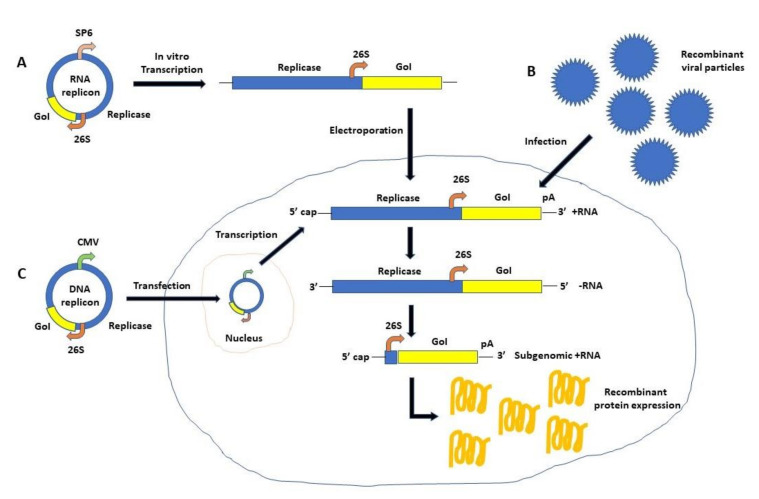
**Schematic illustration of self-replicating RNA alphavirus-based expression systems.** Alphavirus-based delivery and expression systems comprise of infection of recombinant viral particles, electroporation/lipid-based transfection of in vitro transcribed RNA or transfection of plasmid DNA. Recombinant protein expression can be obtained as follows. In vitro transcribed RNA carrying the replicase gene and the gene of interest is electroporated/transfected into mammalian host cells (**A**). Alternatively, the replicon RNA can be delivered to host cells by infection with recombinant alphavirus particles (**B**). The third option is to transfect alphavirus DNA replicons (**C**), which after DNA delivery to the nucleus RNA is in vivo transcribed. The replicase complex will amplify RNA molecules (self-replication) and recombinant protein will be expressed from the 26S subgenomic promoter. 5′ cap, 5′ end cap analogue; 26S, alphavirus subgenomic promoter; CMV, cytomegalovirus promoter; GoI, gene of interest; pA, poly A signal; SP6, bacteriophage SP6 RNA polymerase promoter.

**Figure 2 vaccines-09-01187-f002:**
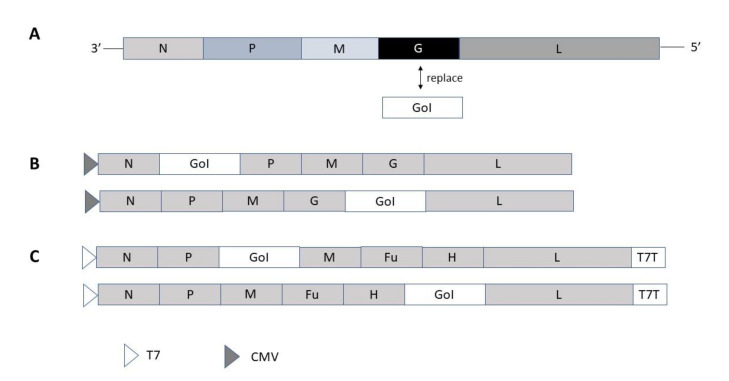
**Self-replicating RNA viral vectors of negative polarity**. (**A**) VSV vector for replacement of VSV G protein. (**B**) Rabies virus and (**C**). Measles virus expression vectors. CMV, cytomegalovirus promoter; Fu, fusion protein, G, glycoprotein; GoI, gene of interest; H, hemagglutinin; L, large protein; M, matrix protein; P. phosphoprotein; T7, T7 RNA polymerase promoter; T7T, T7 terminal sequence.

**Figure 3 vaccines-09-01187-f003:**
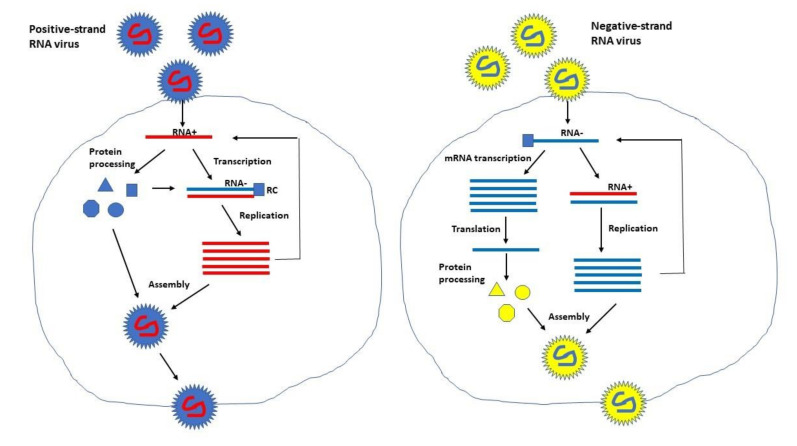
**Replication strategy of positive- and negative-strand self-replicating RNA viruses.** For positive-strand RNA viruses, the viral RNA is directly translated in the cytoplasm and replication of new positive-strand RNA copies require transcription of a negative-strand RNA template. In contrast, negative-strand RNA viruses rely on mRNA transcription before translation can take place.

**Table 1 vaccines-09-01187-t001:** Examples of preclinical studies on self-replicating RNA viral vector vaccines against infectious diseases.

Virus/Disease	Antigen	Vector	Findings	Ref.
**Alphaviruses**				
CHIKV/CHIK	E3-E2-6K-E1	Chimeric VSV-Env	Protection against CHIKV in mice	[32]
VEEV/VEE	E3-E2-6K-E1	VEEV-Env	Protection against VEE in mice, macaques	[33]
WEEV/WEE	E3-E2-6K-E1	WEEV-Env	Partial protection in macaques, strong in mice	[33]
EEEV/EEE	E3-E2-6K-E1	EEEV-Env	Protection against EEE in mice, macaques	[33]
VEEV/VEE	V4020 strain	VEEV DNA	Protection against VEE in mice	[34]
VEEV/VEE	V4020 strain	VEEV DNA	Protection against VEE in macaques	[35]
CHIKV/CHIK	C, Env	MV-CHIKV VLPs	Protection against CHIKV in macaques	[36]
**Arenaviruses**				
LASV/LHF	GPC	VSV-GPC	Protection in guinea pigs and macaques	[37]
LASV/LHF	GPC	MV-LASV-GPC	Protection against LASV in macaques	[38]
LASV/LHF	GPC	YFV-LASV GPC	80% protection in guinea pigs, vector instability	[39]
LASV/LHF	GPC G1/G2	YFV-LASV G1 + G2	83% protection in guinea pigs, stable vector	[40]
LASV/LHF	GPC G1/G2	YFV-LASV G1 + G2	No protection in marmosets	[41]
LASV/LHF	GPC or NP	VEEV-GPC/NP	Protection in guinea pigs after 3 immunizations	[42]
LASV/LHF	GPC	Multivalent VEEV	Protection in inbred CBA/J mice	[43]
JUNV/AHF	GPC	VEEV-GPC	Protection against JUNV in mice	[44]
MACV/BHF	GPC	VEEV-GPC	Protection against MACV in mice	[44]
**Filoviruses**				
EBOV/EVD	GP D637L	KUN-GP D637L	Protection in 75% of nonhuman primates	[45]
EBOV/EVD	GP	VSV-GP	Protection against two EBOV strain in macaques	[46,47]
MARV/MHF	GP	VSV-GP	Protection against MARV in macaques	[48]
SUDV/EVD	GP	VEEV-GP	Protection against SUDV and EBOV in macaques	[48]
**Flaviviruses**				
DENV/DF	E85	VEEV-E85	Protection against DENV in mice	[49]
DENV/DF	ED3	MV-ED3	Strong immunogenicity, partial protection in mice	[50]
DENV/DF	Tetravalent DENV	YFV (CYD-TDV)	Good safety, immunogenicity in rodents, primates	[51,52]
DENV/DF	Tetravalent DENV	YFV (CYD-TDV)	Approved vaccine for endemic populations	[53]
ZIKV/ZVD	prME	VEEV-NLC RNA	Protection in mice with 10 ng of RNA	[54]
ZIKV/ZVD	prME	Chimeric VSV-prME	Protection against ZIKV in mice	[32]
ZIKV/ZVD	E-NS1	VSV-E-NS1	Protection against ZIKV in mice	[55]
**Hepatotropic**				
HBV/Hepatitis	HBsAg	MV-HBsAg	Protection against HBV in 50% of rhesus monkeys	[56]
HBV/Hepatitis	MHB	SFV-MHB	Protection against HBV in mice	[57]
HBV/Hepatitis	HBcAg	SFV-HBcAg	No protection against HBV in mice	[57]
**Lentiviruses**				
HIV/AIDS	HIV gp160 Env	MV-gp160 Env	Humoral and cellular immune responses in mice	[58]
HIV/AIDS	HIV Env	SFV-Env	Superior immunogenicity compared to immunization with DNA and Env protein	[59]
HIV/AIDS	HIV Env	SFV-Env RNA	Immune response in 75% of mice	[60]
HIV/AIDS	HIV Env/Gag/PolRT	SFV RPs/RNA	VLPs superior immunogenicity to RNA in mice	[61]
HIV/AIDS	HIV Env, Gag/Pol/Nef	SFV DNA	Robust immune responses in mice	[62]
HIV/AIDS	HIV gp140	VEEV-RNA-CNE	Superior Ab response compared to VLPS in primates	[63]
**Influenza Viruses**				
IFVA/Influenza	HA	MV AIK-C-HA	Protection against influenza virus in cotton rats	[64]
IFVA/Influenza	HA, NA	VSVΔG-HA/NA	Protection against influenza virus in mice	[65]
IFVA/Influenza	HAfl	VSV-HAfl	Protection against influenza virus in mice	[66]
IFVA/Influenza	HA	VEEV-HA	Protection in chickens	[67]
IFVA/Influenza	HA	SFV-HA RNA	Protection in 90% of mice	[68]
IFVA/Influenza	HA	VEEV-HA RNA	Protection in mice with 64-fold less RNA *	[69]
IFVA/Influenza	M2e	SIN E2S1-M2e	Protection in mice	[70]
IFVA/Influenza	HA, NP	CSFV-HA/NP VRPs	Strong humoral and cellular response in pigs	[71]
**Coronaviruses**				
SARS-CoV/SARS	S	VEEV-S	Protection against SARS-CoV in mice	[72]
MERSCoV/MERS	S	VSVΔG-S	Neutralizing Abs and T cell responses in monkeys	[73]
SARS-CoV-2/COV	S	MV-S	Th1-biased Ab and T cell responses in mice	[74]
SARS-CoV-2/COV	S	MV (TMV-083)	Phase I: weak immunogenicity, trial discontinued	[75,76]
SARS-CoV-2/COV	S	VSV-S	Neutralizing Abs, protection in mice	[77]
SARS-CoV-2/COV	S	VSV (V590)	Phase I: weak immunogenicity, trial discontinued	[78,79]
SARS-CoV-2/COV	S	VSVΔG-S	Protection against SARS-CoV-2 in hamsters	[80]
SARS-CoV-2/COV	S	VSVΔG-S	Phase I/II: study in progress	[81,82]
PEDV/PED	S fragment	BVDV	Neutralization of BVDV and PEDV in mice	[83]
SARS-CoV-2/COV	S	LNP-VEEV-S RNA	Robust Ab responses in mice	[84]
SARS-CoV-2/COV	S	LNP-VEEV-S RNA	Phase I/II: study in progress	[85]
SARS-CoV-2/COV	S	LUNAR-VEEV RNA	Protection in mice after single dose	[86]
SARS-CoV-2/COV	S	SIN-S + αOX40	Protection against SARS-CoV-2 in mice	[87]
**Bacterial**				
*B. anthracis*/Anthrax	PA	SIN-PA	Immune responses, some protection in mice	[88]
*B. abortus*/Brucellosis	*B. abortus* IF3	SFV-CS	Immune responses, protection in mice	[89]
**Parasitic**				
Plasmodium/Malaria	Pf332 antigen	SFV-Pf332	Robust Th1-type immune response in mice	[90]
Plasmodium/Malaria	*P. yoelii* CS epitope	SIN-CS	Protection against malaria in mice	[91]
Leishmania/Leishmaniasis	PpSP15-LmSTI1	SFV-PpSP15-LmSTI1	Superior expression from replicon RNA	[92]

αOX40, immunostimulatory antibody; Ab, antibody; AHF, Argentine hemorrhagic fever; BHF, Bolivian hemorrhagic fever; BVDV, bovine viral diarrhea virus; C, capsid, CHIKV, Chikungunya virus; CNE, cationic nanoemulsion; COV, COVID-19 CS, circumsporozoite; DENV, Dengue virus; DF, Dengue fever; EBOV, Ebola virus; EEEV, eastern encephalitis virus; E85, DENV envelope ectodomain; ED3, DENV envelope protein domain III; Env, envelope proteins; EVD, Ebola virus disease; G1/G2; Glycoprotein subunit of GPC; GPC, glycoprotein complex; HA, hemagglutinin; HAfl, full-length HA; HBcAg, HBV core antigen; HBsAg, HBV surface antigen; HBV, hepatitis B virus; IFVA, influenza virus A; JUNV, Junin virus; LASV, Lassa virus; LHF, Lassa hemorrhagic fever; MACV, Machupo virus; MARV, Marburg virus; MERS-CoV, Middle East respiratory syndrome-coronavirus; MHB, middle HBV surface envelope glycoprotein; MHF, Marburg hemorrhagic fever; NA, neuraminidase; NP, nucleoprotein; NS1, nonstructural protein 1; PA, protective antigen; PEDV, porcine epidemic diarrhea virus; prME, membrane-envelope protein; RPs, recombinant particles; RSP, recombinant subviral particle; S, spike protein; SFV, Semliki Forest virus; VEEV, Venezuelan equine encephalitis virus; VLPs, virus-like particles; VSV, vesicular stomatitis virus; WEEV, western equine encephalitis virus; YFV, yellow fever virus; ZIKV, Zika virus; ZVD, Zika virus disease.* Compared to synthetic mRNA.

**Table 3 vaccines-09-01187-t003:** Examples of preclinical studies on self-replicating RNA viral vector vaccines against cancers.

Cancer	Antigen/Therapeutic	Vector	Findings	Ref.
**Brain**				
Glioblastoma	Endostatin	SFV	Complete tumor regression in mice	[119]
Glioblastoma	IL-18	DC-SFV + IL-12	Enhanced Th1-type response, anti-tumor immunity	[120]
Glioblastoma	gp100, IL-18	SIN DNA	Therapeutic effect, prolonged survival in mice	[121]
Glioblastoma	CHIKV E3-E2-6K-E1	VSVΔG-CHIKV	Selective infection, elimination of tumor cells	[122]
Glioblastoma	GFP, CEA, NIS	GSC-MV	Anti-tumor effect, prolonged survival in mice	[123]
Glioblastoma	EGFP	SFV VA	Tumor inhibition, prolonged survival in mice	[124]
CT-2A glioma	miRT124	SFV4	Replication in tumor cells, prolonged survival	[125]
**Breast**				
A2L2	HER2/neu	Ad/SIN DNA	Prolonged survival in mice	[126]
A2L2	HER2/neu	Ad + SIN DNA	Tumor protection in mice with 80% less DNA	[127]
HER2	HER2 ECD, TMs	VEEV (VRP-HER2)	Preventive, therapeutic tumor growth control in mice	[128]
4T1	IL-12	SFV + *S. typhimurium*	Inhibition of metastasis, long-term survival in mice	[129]
TNBC	M1	M1 + Doxorubicin	Synergistic effect of M1 and Doxorubicin	[118]
MCF7	SLAMblind	MV	Targeting and killing of breast cancer cells	[130]
**Cervical**				
HPV-16	Capsid	MV	Humoral immune responses in mice	[131]
HPV-16	Capsid	MV + HPV protein	IgG and neutralizing antibody responses	[132]
CRPV	E1, E2, E6, E7	VSV	Reduced papilloma volumes, elimination of disease	[133]
HPV-16	E7	VSV	Tumor regression in mice	[134]
HPV-16	E7	VEEV	Immune response, protection against tumors in mice	[135]
HPV-16	E6/E7 fusion	SFVEnh	Tumor regression, complete eradication	[136]
HPV	E6-E7	SFV DNA + EP	85% of immunized mice became tumor-free	[137]
**Colon**				
CT26	GM-CSF	KUN	Tumor regression, cure in 50% mice	[138]
CT26	VEGFR-2	SFV	Inhibition of tumor growth, metastasis	[139]
CT26	VEGFR-2 + IL-4	SFV	Super immunogenicity, prolonged survival	[139]
CT26	LacZ	SFV RNA	Tumor regression, protection against tumor cells	[140]
MC28cea	GM-CSF	MV	Tumor regression, prevention of re-engraftment	[141]
**Lung**				
H358cea	EGFP	SFV	Protection against HBV in 50% of rhesus monkeys	[142]
A549	EGFP	SFV VA	Superior survival compared to adenovirus delivery	[143]
CT26	LacZ	SIN	Complete tumor remission, prolonged survival	[144]
CL25	oMV	MV Hu-191	Suppressed tumor growth, prolonged survival	[145]
LLC	oMV	MV Schwarz	Suppression of uncontrollable tumor growth	[146]
Adenocarcinoma	CEA	MV	Tumor regression in mice	[147]
H2009, A549	IFNβ	VSV	Tumor regression in mice	[148]
LM2	IFNβ	VSV	Prolonged survival, cure in 30% of mice	[148]
**Melanoma**				
B16-OVA	GM-CSF	KUN	Tumor regression, cure of more than 50% of mice	[138]
B16-OVA, B16F0	SIIINFEKL epitope	YFV	Immune response, protection in mice	[149]
B16	TRP-2	VEEV	Immune response, prolonged survival in mice	[150]
B16	TRP-2	VEEV + GITR mAb	Complete tumor regression in 90% of mice	[151]
B16	TRP-2	VEEV + CTLA-4 mAb	Complete tumor regression in 50% of mice	[151]
B16	VEGFR-2/IL-12 + survivin/β-hCG	SFV DNA	Superior tumor growth inhibition, prolonged survival after combination therapy	[152]
mel Z	oMV	MV L-16	Tumor cell killing, inhibition of tumor growth	[153]
B16-OVA	LCMV GP	VSV	Efficacy in subcutaneous tumor models	[154]
**Ovarian**				
A2780	LCMV GP	VSV + ruxolitinib	Tumor regression in mice	[155]
SKOV3ip.1	αFR scFV	MV	Tumor volume reduction, prolonged survival	[156]
SKOV3ip.1	CEA, NIS	MV	Dual therapy superior in mice	[157]
ES2	IL-12	SIN + irinotecan	Long-term survival in mice	[158]
MOSEC	OVA	SFV + VV	Immune response, enhanced anti-tumor activity	[159]
**Pancreatic**				
PDAC	GFP	VSV	Superior oncolytic activity compared to Sendai, RSV	[160]
PDAC	GFP	VSV-ΔM51	Anti-tumor activity enhanced by gemcitabine	[161]
KLM1	SLAMBlind	MV	Suppression of tumor growth in mice	[162]
Capan-2	SLAMBlind	MV	Suppression of tumor growth in mice	[162]
**Prostate**				
PC-3	CEA	MV	Delay in tumor growth, prolonged survival in mice	[163]
PC-3	oMv, oMuV	MV + MuV	Immune responses, protection in mice	[164]
DU145, PC-3	GFP	VSV-ΔM51	Apoptosis in tumor cells, prolonged survival	[165]
DU-145, 22Rv1	LCMV GP	VSV	Long-term remission in mice	[166]
TRAMP-C	PSMA	VEEV	PSMA-specific immune response in mice	[167]
TRAMP	STEAP	VEEV	STEAP-specific immune response, prolonged survival	[168]
TRAMP	PSCA	VEEV	Long-term survival for 12 months in 90% of mice	[169]

Ad, adenovirus; CEA, carcinoembryonic antigen; CRPV, cottontail rabbit papilloma virus; CSC, glioma stem cell; CTLA-4, CTL antigen-4; DC, dendritic cell; GFP, green fluorescent protein; GITR, glucocorticoid-induced tumor necrosis factor receptor; GM-CSF, granulocyte macrophage-colony stimulating factor; IL. interleukin; KUN, Kunjin virus; LCMV, lymphocytic choriomeningitis virus; Lewis lung carcinoma; MOSEC, murine ovarian surface epithelial carcinoma; MV, measles virus; NIS, sodium iodine symporter; PDAC, pancreatic ductal adenocarcinoma; RROC, refractory recurrent ovary cancer; RSV, respiratory syncytial virus; SFV, Semliki Forest virus; TNBC, triple-negative breast cancer; SIN, Sindbis virus; STEAP, six-transmembrane epithelial antigen of the protein; TRAMP, transgenic adenocarcinoma mouse prostate; VEEV, Venezuelan equine encephalitis virus; VEGFR-2, vascular endothelial growth factor receptor-2; VSV, vesicular stomatitis virus; YFV, yellow fever virus.

**Table 4 vaccines-09-01187-t004:** Examples of clinical studies on self-replicating RNA viral vector vaccines against cancers.

Cancer	Antigen/Therapeutic	Vector	Findings	Ref.
**Breast**				
HER2	HER2 ECD TMs	VEEV (VRP-HER2)	Phase I: immune response, PR and SD	[128]
HER2	HER2 ECD TMs	VEEV (VRP-HER2)	Phase II: study in progress	[170]
**Cervical**				
HPV-16	E6/E7 fusion	SFVEnh (Vvax001)	Phase I: safe, immune responses in all patients	[171]
**Colon**				
Stage III-IV	CEA	VEEV	Phase I: immune responses, prolonged survival	[172]
**Ovarian**				
RROC	CEA	MV	Phase I: well-tolerated, dose-dependent activity	[173]
**Pancreatic**				
Metastatic	CEA	VEEV	Phase I: immunogenicity, prolonged overall survival	[174]

CEA, carcinoembryonic antigen; MV, measles virus; RROC, refractory recurrent ovary cancer; VEEV, Venezuelan equine encephalitis virus.

## Data Availability

This is a review, so no experimental data are available.

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
