# Peer review of "Self-Replicating RNA Viruses for Vaccine Development against Infectious Diseases and Cancer"

_vaccines, 2021, doi:10.3390/vaccines9101187_

Round 1

Reviewer 1 Report

The resubmitted review article (vaccines-1363620, originally vaccines-1314153) has been slightly improved, however, most of my concerns are still not addressed. In addition, a newly added figure 2 is unacceptable and modified tables are still not organized. Further, newly added sentences as well as modified sentences still contains many errors as well as old information. Therefore, unfortunately, the article is still not sufficient for Vaccines level.

One critical problem could be that the article, which shows vaccine approaches using different gene delivery systems (replicons, replicon particles, single-cycle and replication-complement recombinant virus vectors) of RNA viruses with different replication cycles, was written mainly based on Alphavirus replicon and gene delivery systems. Therefore, there are many misunderstandings and contradictions, which could mislead the readers to wrong conclusions. It would be better to describe different virus vectors separately to avoid further confusions.

The reviewer totally agrees that the concept of "self-replicating/amplifying RNA" was first developed based on the Alphavirus replication mechanism. Therefore, initially, and perhaps even now, the term "self-replicating/amplifying RNA" means "non-segmented viral RNA genomes amplified by the replicase complex, whose components have to be translated directly from them". In case of negative-stranded RNA viruses, the RNA genomes are never replicated and translated until additional components, including RNA-dependent RNA polymerase (RdRp), are provided in trans and anti-genomic (positive-stranded) RNAs are synthesized. Therefore, negative-stranded RNA virus genomes are not self-amplified. Of course, once the virus particles are generated, the RNA genomes keep replicating in the infected host cells. The author tries to define "self-replicating/amplifying RNA viruses", which include viruses belonged to the family Flaviviridae and the genus Alphavirus as well as some viruses of the order Mononegavirales. It is slightly hard to agree with his extended interpretation of "self-replicating/amplifying". For example, it is absolutely obvious that almost all the viruses, regardless of classification, can replicate in the susceptive cells. Thus, they are all a kind of "self-replicating viruses"??

The author keeps using similar figures to explain alphavirus gene delivery systems in his review articles. The figures in his/his groups' previous articles (for example, doi: 10.2174/1381612823666170622094715) is much more organized than Figure 1 in this article.

Figure 2 is not acceptable. The figure only shows incomplete genomic constructs of the viruses. Readers, who are unfamiliar to the negative-stranded RNA viruses, cannot understand what the figure indicates. It would be better to show the replication/gene expression mechanism of the negative-stranded RNA virus replicon to make the readers clearly understand why some of the negative-stranded RNA viruses are "self-replicating/amplifying RNA viruses".

The flavivirus replicon systems, which are also different from alphavirus systems, should be described properly.

Page 1 lanes 41 and 42: Although the author wrote " The advantages and disadvantages of using self-replicating RNA viral vectors, especially RNA-based delivery, is also discussed.", he mentioned only a little in the text (lanes 670-674?).

Tables are still disorganized. it would be better to clearly show which gene delivery systems are used.

Page 11 lanes 403 and 404 "Although...here.": Only the vaccines for cervical cancer are preventive/prophylactic vaccines. Most of the listed vaccines are virotherapy. Please consider the sentence.

The article is still just a list of roughly summarized data. More interpretation must be required. The reviewer wonders if the readers do care about how much viruses or RNA/DNA were injected to the mouse, which cell lines were used... and so on? It would be arduous for them to read a long and slightly rambling list of previous reports.

There are too many errors and insufficient and old information. Some of them (not all) are listed below. It would be better for the author to take much more time to go through a whole manuscript carefully.

  1. Page 2 lane 48 and 49 "RNA viruses deposited....DNA-based delivery.": Not true. There are many RNA viruses, whose RNA genomes are delivered to the nuclei.
  2. Page 2 lanes 61-63 "Moreover,....host genome.": Unclear phrase. "RNA transcribed from recombinant particles and RNA replicons is degraded within 5-7 days post-immunization" so.....?
  3. Page 2 lanes 89-91 "In the case...self-replication": Not true. RdRp is encoded in L gene and P (phosphoprotein) protein is an essential cofactor for the RdRp activity.
  4. Fig. 2: Fig. 2B should be the genome constructs of Measles virus and Fig. 2C should be Rabies virus genome constructs. In addition, there are much more variations of the VSV genome constructs.
  5. Page 2 lanes 86-88: Old information. Generation of CSFV-based vaccine for influenza was reported.
  6. Page 4 lanes 114-123: The authors should first describe "what recombinant vaccinia virus vectors deliver". The reviewer could realize that the bacteriophage T7 polymerase gene is delivered by recombinant vaccinia virus to initiate the transcription of virus genomic/anti-genomic RNA under control of T7 promoter. However, the readers, who are not from the field, cannot understand it. As vaccinia virus, which is DNA virus, suddenly appears in the descriptions regarding VSV and other negative-stranded RNA virus vectors, the readers must be confused. In addition, "In many cases, application of reverse genetics for expression vector engineering has been based on recombinant vaccinia virus vectors." is not true recently. As the authors also describes in the following text, these 20 years, vaccinia-free systems became major. The sentence should be amended.

Author Response

The resubmitted review article (vaccines-1363620, originally vaccines-1314153) has been slightly improved, however, most of my concerns are still not addressed. In addition, a newly added figure 2 is unacceptable and modified tables are still not organized. Further, newly added sentences as well as modified sentences still contains many errors as well as old information. Therefore, unfortunately, the article is still not sufficient for Vaccines level.

 Response: I sincerely appreciate the thorough reviewing carried out by the reviewer. I have carefully reviewed the comments and have addressed the issues where I have seen it necessary and/or justified my view when respectfully disagreeing with the requests.

One critical problem could be that the article, which shows vaccine approaches using different gene delivery systems (replicons, replicon particles, single-cycle and replication-complement recombinant virus vectors) of RNA viruses with different replication cycles, was written mainly based on Alphavirus replicon and gene delivery systems. Therefore, there are many misunderstandings and contradictions, which could mislead the readers to wrong conclusions. It would be better to describe different virus vectors separately to avoid further confusions.

Response: Attempts have been made to correct any misunderstandings and contradictions.

The reviewer totally agrees that the concept of "self-replicating/amplifying RNA" was first developed based on the Alphavirus replication mechanism. Therefore, initially, and perhaps even now, the term "self-replicating/amplifying RNA" means "non-segmented viral RNA genomes amplified by the replicase complex, whose components have to be translated directly from them". In case of negative-stranded RNA viruses, the RNA genomes are never replicated and translated until additional components, including RNA-dependent RNA polymerase (RdRp), are provided in trans and anti-genomic (positive-stranded) RNAs are synthesized. Therefore, negative-stranded RNA virus genomes are not self-amplified. Of course, once the virus particles are generated, the RNA genomes keep replicating in the infected host cells. The author tries to define "self-replicating/amplifying RNA viruses", which include viruses belonged to the family Flaviviridae and the genus Alphavirus as well as some viruses of the order Mononegavirales. It is slightly hard to agree with his extended interpretation of "self-replicating/amplifying". For example, it is absolutely obvious that almost all the viruses, regardless of classification, can replicate in the susceptive cells. Thus, they are all a kind of "self-replicating viruses"??

Response: The description of positive- and negative-strand RNA viruses has been updated and clarified by the addition of Figure 3. I appreciate the reviewer’s view on negative-strand RNA viruses as not fulfilling the criteria for self-replicating RNA viruses. However, it seems that other scientists share my view of “extending self-replication to negative strand RNA viruses” as I have been invited to write a review titled “Self-replicating vehicles based on negative strand RNA viruses” for Cancer Gene Therapy.

The author keeps using similar figures to explain alphavirus gene delivery systems in his review articles. The figures in his/his groups' previous articles (for example, doi: 10.2174/1381612823666170622094715) is much more organized than Figure 1 in this article.

Response: Figure 1 has been modified to be as “organized” as in the publication the reviewer referred to.

Figure 2 is not acceptable. The figure only shows incomplete genomic constructs of the viruses. Readers, who are unfamiliar to the negative-stranded RNA viruses, cannot understand what the figure indicates. It would be better to show the replication/gene expression mechanism of the negative-stranded RNA virus replicon to make the readers clearly understand why some of the negative-stranded RNA viruses are "self-replicating/amplifying RNA viruses".

Response: Figure 2 has been updated and in support Figure 3 has been added.

The flavivirus replicon systems, which are also different from alphavirus systems, should be described properly.

Response: The Kunjin virus expression system is described as a member of flaviviruses.

Page 1 lanes 41 and 42: Although the author wrote " The advantages and disadvantages of using self-replicating RNA viral vectors, especially RNA-based delivery, is also discussed.", he mentioned only a little in the text (lanes 670-674?).

Response: In fact, the description starts on line 667. Anyway, the text has been expanded.

Tables are still disorganized. it would be better to clearly show which gene delivery systems are used.

Response: I respectfully disagree. The way the tables are organized is according to disease indications. If it is based on delivery systems, there would be a lot of repetition and at least I feel the tables would be more difficult to read.

Page 11 lanes 403 and 404 "Although...here.": Only the vaccines for cervical cancer are preventive/prophylactic vaccines. Most of the listed vaccines are virotherapy. Please consider the sentence.

Response: I respectfully disagree with this comment as there are several examples (other than cervical cancer) given in the text and Table 2 of protection against challenges with tumor cells. However, “prophylactic and therapeutic” has been added in front of “cancer vaccines”.

The article is still just a list of roughly summarized data. More interpretation must be required. The reviewer wonders if the readers do care about how much viruses or RNA/DNA were injected to the mouse, which cell lines were used... and so on? It would be arduous for them to read a long and slightly rambling list of previous reports.

Response: With all my respect, I have been requested by other reviewers for other manuscripts to explicitly include information on virus/RNA/DNA doses. Moreover, the point to mention the RNA and DNA doses is to demonstrate that with self-replicating/amplifying RNA doses can be reduced to cause less harm on normal tissue and increase the vaccination capacity or alternatively with higher doses potentially generate enhanced immune responses. Related to the comment that “the article is still just a list of roughly summarized data”, by definition, “a review article is an article that summarizes the current state of understanding on a topic. A review article surveys and summarizes previously published studies, instead of reporting new facts or analysis” (Wikipedia).

There are too many errors and insufficient and old information. Some of them (not all) are listed below. It would be better for the author to take much more time to go through a whole manuscript carefully.

Response: Major efforts (see below) have been made to correct errors and insufficient information. Concerning “old information”, I believe that citing classic technology development and pioneering and breakthrough findings is necessary. Every effort has been made to include the latest available information as indicated by the many references from 2020 and 2021 and references on clinical trials in progress.

  1. Page 2 lane 48 and 49 "RNA viruses deposited....DNA-based delivery.": Not true. There are many RNA viruses, whose RNA genomes are delivered to the nuclei.

Response: To specify, “self-replicating” has been added before RNA viruses” and the delivery of influenza virus RNA to the nucleus is mentioned.

  1. Page 2 lanes 61-63 "Moreover,....host genome.": Unclear phrase. "RNA transcribed from recombinant particles and RNA replicons is degraded within 5-7 days post-immunization" so.....?

Response: The text has been modified to better describe the sensitivity of RNA to degradation and the lack of reverse transcription and integration into the host genome.

  1. Page 2 lanes 89-91 "In the case...self-replication": Not true. RdRp is encoded in L gene and P (phosphoprotein) protein is an essential cofactor for the RdRp activity.

Response: Thank you for pointing it out! The text has been revised accordingly.

  1. Fig. 2: Fig. 2B should be the genome constructs of Measles virus and Fig. 2C should be Rabies virus genome constructs. In addition, there are much more variations of the VSV genome constructs.

Response: The positions of measles virus and rabies virus have been switched in Fig. 2 as it makes sense too present rhabdoviruses (VSV and rabies) together. Additional information on VSV vector constructs has been included.

  1. Page 2 lanes 86-88: Old information. Generation of CSFV-based vaccine for influenza was reported.

Response: As was the case for the first revision, I have searched both PubMed and Google using “CSFV-based vaccines”, “Classic swine fever virus-based influenza vaccines”, “Recombinant CSFV vaccines” etc and cannot find a single publication on the topic. If the reviewer kindly can provide the reference(s), I would be happy to include it (them) in the manuscript.

  1. Page 4 lanes 114-123: The authors should first describe "what recombinant vaccinia virus vectors deliver". The reviewer could realize that the bacteriophage T7 polymerase gene is delivered by recombinant vaccinia virus to initiate the transcription of virus genomic/anti-genomic RNA under control of T7 promoter. However, the readers, who are not from the field, cannot understand it. As vaccinia virus, which is DNA virus, suddenly appears in the descriptions regarding VSV and other negative-stranded RNA virus vectors, the readers must be confused. In addition, "In many cases, application of reverse genetics for expression vector engineering has been based on recombinant vaccinia virus vectors." is not true recently. As the authors also describes in the following text, these 20 years, vaccinia-free systems became major. The sentence should be amended.

Response: The text has been revised pointing out that originally vaccinia virus vectors were used and also explaining the presence of the T7 polymerase gene.

Reviewer 2 Report

In the review „Self-replicating RNA Viruses for Vaccine Development against Infectious Diseases and Cancer” of Kenneth Lundstrom, the author provided a carefully collected summary on RNA viruses for vaccine development.

It is the opinion of this reviewer, that the presented review represents and outstanding manuscript. The review is well written, comprises comprehensively collected information that are presented is a very structured way to the readership. Furthermore, the manuscript is adequate in style and length and comprises almost no spelling errors.

Well done!

The is only a very minor comment – please also adapt the style of the bacterial designations in the references section to italics.

I have no further comments to improve the manuscript!

Author Response

In the review „Self-replicating RNA Viruses for Vaccine Development against Infectious Diseases and Cancer” of Kenneth Lundstrom, the author provided a carefully collected summary on RNA viruses for vaccine development.

It is the opinion of this reviewer, that the presented review represents and outstanding manuscript. The review is well written, comprises comprehensively collected information that are presented is a very structured way to the readership. Furthermore, the manuscript is adequate in style and length and comprises almost no spelling errors.

Well done!

The is only a very minor comment – please also adapt the style of the bacterial designations in the references section to italics.

Response: I am pleased to hear that the reviewer liked the review and appreciate his/her kind words. The bacterial and protozoan names have been revised to italics in the reference section now.

Reviewer 3 Report

This paper by Lundstrom presents an updated review of the use of self-replicating RNA viruses for vaccine development against infectious diseases and cancer.

The review is comprehensive and covers most of the relevant issues. There are however several issues that would benefit from additional elaboration:

1) The use of MV-based vaccines may face the issue of immunity in the human population. Studies done in mice do not allow to assess whether pre-existing immunity, due to vaccination, in human populations could compromise vaccine efficacy.

2) Tables 1 and 2: it would be helpful to separate into a different table studies done in humans.

3) The use of live-attenuated, non-cytolytic, virus-based vaccines for active immunotherapy should be incorporated into the review.

4) The section on oncolytic viruses should incorporate a discussion about the differences between tumor and healthy cells at the root of the efficacy of oncolytic viruses in the absence of spread virus induced cellular and tissue damage.

Author Response

This paper by Lundstrom presents an updated review of the use of self-replicating RNA viruses for vaccine development against infectious diseases and cancer.

The review is comprehensive and covers most of the relevant issues. There are however several issues that would benefit from additional elaboration:

Response: I am grateful to the nice comments by the reviewer.

1) The use of MV-based vaccines may face the issue of immunity in the human population. Studies done in mice do not allow to assess whether pre-existing immunity, due to vaccination, in human populations could compromise vaccine efficacy.

Response: Although briefly mentioned (lines 519-520), this important point has been added to the Conclusions section.

2) Tables 1 and 2: it would be helpful to separate into a different table studies done in humans.

Response: The preclinical and clinical examples are now presented in different tables; Preclinical data in Tables 1 and 3 and clinical findings in Tables 2 and 4.

3) The use of live-attenuated, non-cytolytic, virus-based vaccines for active immunotherapy should be incorporated into the review.

Response: Live-attenuated virus-based vaccines are described (see references 61, 62, 152). However, as many of the live-attenuated virus vaccines are based on influenza viruses and other not self-replicating RNA viruses, their presentation is limited here.

4) The section on oncolytic viruses should incorporate a discussion about the differences between tumor and healthy cells at the root of the efficacy of oncolytic viruses in the absence of spread virus induced cellular and tissue damage.

Response: A sentence has been added describing the replication in and killing of tumor cells by oncolytic viruses without causing damage to normal cells (lines 428-430).

Reviewer 4 Report

Manuscript ID: vaccines-1363620

This manuscript describes a well performed review on self-amplifying RNA viruses as potential vaccines against viral infections and tumors. Various vaccination strategies and effectiveness against many viral and tumor challenge models have been clearly documented. Especially the current COVID-19 pandemic and the speed of generating effective RNA vaccines against SARS-CoV-2 has clearly set the spot lines on this platform. The review is acceptable for publication.

The format of Table 1 needs attention:

  • Continue the headings of the table (Virus/Disease --- Antigen --- Vector --- Findings --- Ref) on every page where the table is printed.
  • The rows in the different cells in the Table do not match. Using a smaller letter font will overcome this problem.

Author Response

This manuscript describes a well performed review on self-amplifying RNA viruses as potential vaccines against viral infections and tumors. Various vaccination strategies and effectiveness against many viral and tumor challenge models have been clearly documented. Especially the current COVID-19 pandemic and the speed of generating effective RNA vaccines against SARS-CoV-2 has clearly set the spot lines on this platform. The review is acceptable for publication.

Response: Many thanks for your nice comments! 

The format of Table 1 needs attention:

  • Continue the headings of the table (Virus/Disease --- Antigen --- Vector --- Findings --- Ref) on every page where the table is printed.

Response: The headings have now been added to the top of each page for Tables 1 and 3 not fitting on one page.

  • The rows in the different cells in the Table do not match. Using a smaller letter font will overcome this problem.

Response: My apologies for the problem with the lines in the tables. This was not the situation in the original tables submitted. Anyway, the lines have now been aligned.

Round 2

Reviewer 1 Report

The author added several sentences and figures to the revised review article (vaccines-1363620, originally vaccines-1314153). However, many of my concerns are still not addressed. In addition, newly added sentences and figures as well as modified sentences contains mistakes and makes other confusions. Further, the manuscript still contains many errors, which could give the readers insufficient and incorrect information. The reviewer understands that the author must have spent a lot of time to make a collection of references regarding ssRNA virus-based gene delivery systems for vaccine developments and cancer therapy and made great efforts to precisely evaluate them, summarize them and write up the manuscript. However, it is slightly hard to say that the manuscript is considerate of the readers. Therefore, unfortunately, the article is still not sufficient for Vaccines level.

The reviewer is now afraid that it is slightly hard to reach an agreement in regard to the definition of "Self-replicating RNA virus". The reviewer's request is the clear explanation regarding what is "Self-replicating/amplifying RNA viruses". The reviewer does not think that it is appropriate to cite comments/invitation of other journals to support the author's view here. Nevertheless, in fact, the reviewer does not feel any problem regarding the phrase "Self-replicating vehicles based on negative strand RNA viruses", too, because viruses are a kind of self-replicating vehicles (as mentioned previous comments). The author first wrote about alphaviruses and explain about original self-replicating/amplifying (replicon) RNA-based gene expression systems using positive-stranded RNA viruses, and then explain other virus (mononegavirus)-based gene expression systems. The reviewer knows that the author first tried to define "Self-replicating RNA virus" in the first part of Chapter 2 (Page 2 lanes 49-51, "The most significant feature of self-replicating RNA viruses relates to their efficient self-replication of RNA from the established RNA replication complex, ..."). However, due to the writing structure of Chapter 2 (and perhaps abstract), it is unclear that "Self-replicating RNA virus" means "the virus, which can self-replicate in the host cells " or "the virus, whose genomic RNA is original self-replicating/amplifying RNA". It would be better to avoid to mislead the readers to think that "Self-replicating RNA viruses" have original self-replicating/amplifying RNA, which means "non-segmented viral RNA genomes amplified by the replicase complex, whose components must be translated directly from them in the cytoplasm".

Another problem could be the unclear reason why the author decides "How far to expand the meaning of "Self-replicating RNA virus"". According to the sentences in lanes 47-52, 65-69, figs, and elsewhere, "Self-replicating RNA virus" is (1) the virus, whose genome must be a single-stranded and non-segmented RNA, (2) the virus, which should replicate in the cytoplasm, (3) the virus, whose genome must be never transferred to nuclei, and (4) the virus, whose genome must be efficiently replicated by viral own RdRp, which could form a replication complex. If the reviewer's understanding is correct, the reviewer wonders why other RNA viruses (segmented-RNA viruses, double-stranded RNA viruses, mononegaviruses whose genomes are transferred to nuclei, and so on) cannot be self-replicating RNA virus, despite they can self-replicate in the host cells and, of course, their RNA genomes are efficiently replicated by their own RdRps?

The reviewer agrees that the review article should give the readers an overview of previous researches on the certain topic. In addition, it must provide a critical and precise evaluation and interpretation of reported data (Clin Biochem 2020 and so on). Further, the review article requires not just a detailed literature search but a thorough 'digest' of the material obtained (sic, Hosp Med 2001). Therefore, showing a list of summarized previous data is not enough. For example, if the author mentioned the RNA and DNA doses in the manuscript to demonstrate that with self-replicating/amplifying RNA doses can be reduced to cause less harm on normal tissue and increase the vaccination capacity or alternatively with higher doses potentially generate enhanced immune responses, he must explain it properly in the text. The author mentioned it only in Chapter 2. The reviewer wonders how the readers can find out that the amounts of nucleotide (or virus) shown in Chapter 3 and 4 of the manuscript are less than usual and indicate the efficiency of self-replicating RNA (or RNA virus)-based gene delivery systems. The reviewer found only a few sentences to describe the efficiency of replicon RNA by comparing the amounts of RNA (for example: replicon vs synthetic mRNA, lanes 303-305).  

The author modified only partially the manuscript by adding/deleting sentences/words. Therefore, some part of manuscript become difficult to understand due to loss of context as well as incorrect description. For example, the sentences in 106-119 contain insufficient information and the duplication of description regarding vaccinia virus-free reverse genetics system. The author first explains vaccinia virus dependent plasmid-based reverse genetics system and mentions that vaccinia virus provides T7 RNA polymerase (T7pol) to initiate RNA transcription under control of T7 promoter. And then he tried to introduce vaccinia virus-free systems, in which the viral RNA constructs (genomic RNA and mRNAs of N, L and P) is still transcribed under control of T7 promoter (and translated N, L and P protein in IRES-dependent manner). However, these sentences are unclear and hard to understand. The vaccinia virus-free systems succeeded not due to T7 promoter nor IRES, but just due to T7pol-expressing cell lines. In addition, the author likely does not want to mention that reverse genetics system for negative-stranded RNA virus always requires to express the components of replicase complex, including RdRp, in trans. The author must describe the basic strategy of reverse genetics system for negative-stranded RNA virus properly but concisely if he wants to introduce it in the manuscript.

There are many parts losing contexts. Please go through a whole manuscript carefully.

The manuscript still contains many incorrect/insufficient descriptions, which mislead the readers. For example, the sentences in lanes 537-542 (and related part of table) could mislead the readers to think that the expression of β-gal encoded in LacZ gene by SFV- LacZ replicon could protect mice from tumor (CT26) challenge and treat existing tumor. In the reference #144, not wild-type CT26 but CT26.CL25, which expresses the model antigen β-gal, was used. Therefore, β-gal cannot be used for treatment of colon cancer (wild-type CT26). The sentences should be amended. The reviewer understands that it could be slightly hard to go through all references carefully one more time, however, the sufficient information must be described properly. Please go thorough the references as much as possible. 

Generation of CSFV-based vaccine for influenza (T Demoulins et al Front Immunol. 2021) was reported early this year (and e-collection last year). Although the research is still in a quite juvenile stage, CSFV-based vaccine development is proceeding step by step. It is very easy to find this paper if you search PubMed using the terms "classical swine fever virus influenza " or "classical swine fever virus self-amplifying RNA ". Even just using "self-amplifying RNA", the paper ranks in the top 20 (currently 17th).    

There a many minor errors, such as wrong usage of terms (for example, "translation" in Fig3), wrong reference numbers (for example, #143 in lane 577 must be #142?), and so on.  

Author Response

The author added several sentences and figures to the revised review article (vaccines-1363620, originally vaccines-1314153). However, many of my concerns are still not addressed. In addition, newly added sentences and figures as well as modified sentences contains mistakes and makes other confusions. Further, the manuscript still contains many errors, which could give the readers insufficient and incorrect information. The reviewer understands that the author must have spent a lot of time to make a collection of references regarding ssRNA virus-based gene delivery systems for vaccine developments and cancer therapy and made great efforts to precisely evaluate them, summarize them and write up the manuscript. However, it is slightly hard to say that the manuscript is considerate of the readers. Therefore, unfortunately, the article is still not sufficient for Vaccines level.

 Response: I am sorry to hear that despite extensive revision you did not find the review suitable for publication despite that three other reviewers stated that “The review is well written, comprises comprehensively collected information that are presented is a very structured way to the readership” (Reviewer 2) and “The review is comprehensive and covers most of the relevant issues” (Reviewer 3), and “The manuscript is well performed review on self-amplifying RNA viruses” (Reviewer 4). On the other hand, I appreciate your dedication and time spent reviewing the manuscript and I sincerely hope that you will find the newly revised version acceptable.

The reviewer is now afraid that it is slightly hard to reach an agreement in regard to the definition of "Self-replicating RNA virus". The reviewer's request is the clear explanation regarding what is "Self-replicating/amplifying RNA viruses". The reviewer does not think that it is appropriate to cite comments/invitation of other journals to support the author's view here. Nevertheless, in fact, the reviewer does not feel any problem regarding the phrase "Self-replicating vehicles based on negative strand RNA viruses", too, because viruses are a kind of self-replicating vehicles (as mentioned previous comments). The author first wrote about alphaviruses and explain about original self-replicating/amplifying (replicon) RNA-based gene expression systems using positive-stranded RNA viruses, and then explain other virus (mononegavirus)-based gene expression systems. The reviewer knows that the author first tried to define "Self-replicating RNA virus" in the first part of Chapter 2 (Page 2 lanes 49-51, "The most significant feature of self-replicating RNA viruses relates to their efficient self-replication of RNA from the established RNA replication complex, ..."). However, due to the writing structure of Chapter 2 (and perhaps abstract), it is unclear that "Self-replicating RNA virus" means "the virus, which can self-replicate in the host cells " or "the virus, whose genomic RNA is original self-replicating/amplifying RNA". It would be better to avoid to mislead the readers to think that "Self-replicating RNA viruses" have original self-replicating/amplifying RNA, which means "non-segmented viral RNA genomes amplified by the replicase complex, whose components must be translated directly from them in the cytoplasm".

Response: I agree that we still disagree on the definition of Self-replicating RNA viruses. However, I have carefully evaluated the contents and have tried my best to come to a productive solution, which should satisfy both parties. For this reason, I have tried to formulate the text on self-replicating RNA viruses to avoid any misleading statements. However, I strongly defend the appropriateness of referring to comments about invitations from other journals/scientists. First, my citation was not in the manuscript, it was only in my response letter. Second, it tells that those other scientists/editors agree with my view on self-replicating viruses. Third, I submitted the manuscript on “Self-replicating Vehicles Based on Negative Strand RNA Viruses” last week to Cancer Gene Therapy.

Another problem could be the unclear reason why the author decides "How far to expand the meaning of "Self-replicating RNA virus"". According to the sentences in lanes 47-52, 65-69, figs, and elsewhere, "Self-replicating RNA virus" is (1) the virus, whose genome must be a single-stranded and non-segmented RNA, (2) the virus, which should replicate in the cytoplasm, (3) the virus, whose genome must be never transferred to nuclei, and (4) the virus, whose genome must be efficiently replicated by viral own RdRp, which could form a replication complex. If the reviewer's understanding is correct, the reviewer wonders why other RNA viruses (segmented-RNA viruses, double-stranded RNA viruses, mononegaviruses whose genomes are transferred to nuclei, and so on) cannot be self-replicating RNA virus, despite they can self-replicate in the host cells and, of course, their RNA genomes are efficiently replicated by their own RdRps?

Response: I feel that the discussion is reaching a philosophical level. Obviously, all viruses replicate, but according to Google asking the question:  Which viruses are self replicating?

“Self-replicating single-stranded RNA viruses such as alphaviruses, flaviviruses, measles viruses, and rhabdoviruses provide efficient delivery and high-level expression of therapeutic genes due to their high capacity of RNA replication”.

The reviewer agrees that the review article should give the readers an overview of previous researches on the certain topic. In addition, it must provide a critical and precise evaluation and interpretation of reported data (Clin Biochem 2020 and so on). Further, the review article requires not just a detailed literature search but a thorough 'digest' of the material obtained (sic, Hosp Med 2001). Therefore, showing a list of summarized previous data is not enough. For example, if the author mentioned the RNA and DNA doses in the manuscript to demonstrate that with self-replicating/amplifying RNA doses can be reduced to cause less harm on normal tissue and increase the vaccination capacity or alternatively with higher doses potentially generate enhanced immune responses, he must explain it properly in the text. The author mentioned it only in Chapter 2. The reviewer wonders how the readers can find out that the amounts of nucleotide (or virus) shown in Chapter 3 and 4 of the manuscript are less than usual and indicate the efficiency of self-replicating RNA (or RNA virus)-based gene delivery systems. The reviewer found only a few sentences to describe the efficiency of replicon RNA by comparing the amounts of RNA (for example: replicon vs synthetic mRNA, lanes 303-305).  

Response: I feel that the comment that the “review article requires not just a detailed literature search but a thorough 'digest' of the material obtained is unjustified. Therefore, showing a list of summarized previous data is not enough” is unfair and not justified. Obviously, Sections 3 and 4 consist of reviewing the achievements with self-replicating RNA viruses for infectious diseases and various cancers. However, the Conclusions Section (lines 698-787) almost 90 lines (!) is a thorough analysis of all aspects including advantages and disadvantages, a digest of findings and achievements, questions on dosing, transition from small animal models though canines to humans, etc, etc.  

I respectfully disagree with the reviewer. The use of lower doses of self-replicating RNA is mentioned in Section 2 (lines 54-55) and in Section 3 (lines 319-321). Moreover, the reduced (80% less) amount of replicon DNA compared to conventional plasmid DNA is described (lines 484-487) and again the need of a 200-fold lower dose (lines 558-560). In contrast to the statement of the reviewer, it is also described in the Conclusions section (lines 708-711). I think this should make it clear to any reader not familiar with self-replicating RNA, but to be absolutely sure, a short statement has been added to the Abstract section (line 14). All these statements on doses of self-replicating RNA are highlighted in blue. Additional text has been added related to the immune stimulation by self-replicating RNA.

The author modified only partially the manuscript by adding/deleting sentences/words.

Response: Is not the definition of modification partial adding/deleting of sentences/words?  

Therefore, some part of manuscript become difficult to understand due to loss of context as well as incorrect description. For example, the sentences in 106-119 contain insufficient information and the duplication of description regarding vaccinia virus-free reverse genetics system. The author first explains vaccinia virus dependent plasmid-based reverse genetics system and mentions that vaccinia virus provides T7 RNA polymerase (T7pol) to initiate RNA transcription under control of T7 promoter. And then he tried to introduce vaccinia virus-free systems, in which the viral RNA constructs (genomic RNA and mRNAs of N, L and P) is still transcribed under control of T7 promoter (and translated N, L and P protein in IRES-dependent manner). However, these sentences are unclear and hard to understand. The vaccinia virus-free systems succeeded not due to T7 promoter nor IRES, but just due to T7pol-expressing cell lines. In addition, the author likely does not want to mention that reverse genetics system for negative-stranded RNA virus always requires to express the components of replicase complex, including RdRp, in trans. The author must describe the basic strategy of reverse genetics system for negative-stranded RNA virus properly but concisely if he wants to introduce it in the manuscript.

Response: I have tried to explain that the first reverse genetics systems were based on vaccinia virus and LATER, there was a shift towards vaccinia-free systems. Every effort has been made to make this clearer and also include as the reviewer rightly pointed out that the T7 polymerase expression was cell line based and that reverse genetics systems for negative strand RNA viruses requires the expression the replicase complex in trans.

There are many parts losing contexts. Please go through a whole manuscript carefully.

The manuscript still contains many incorrect/insufficient descriptions, which mislead the readers. For example, the sentences in lanes 537-542 (and related part of table) could mislead the readers to think that the expression of β-gal encoded in LacZ gene by SFV- LacZ replicon could protect mice from tumor (CT26) challenge and treat existing tumor. In the reference #144, not wild-type CT26 but CT26.CL25, which expresses the model antigen β-gal, was used. Therefore, β-gal cannot be used for treatment of colon cancer (wild-type CT26). The sentences should be amended. The reviewer understands that it could be slightly hard to go through all references carefully one more time, however, the sufficient information must be described properly. Please go thorough the references as much as possible. 

Response: The text has been revised according to the original publication.

Generation of CSFV-based vaccine for influenza (T Demoulins et al Front Immunol. 2021) was reported early this year (and e-collection last year). Although the research is still in a quite juvenile stage, CSFV-based vaccine development is proceeding step by step. It is very easy to find this paper if you search PubMed using the terms "classical swine fever virus influenza " or "classical swine fever virus self-amplifying RNA ". Even just using "self-amplifying RNA", the paper ranks in the top 20 (currently 17th).

Response: Thank you for pointing out the search. The immunization of pigs with CSFV-RepRNA Influenza HA/NP has been described in the text and added to Table 1 (and references).      

There a many minor errors, such as wrong usage of terms (for example, "translation" in Fig3), wrong reference numbers (for example, #143 in lane 577 must be #142?), and so on.  

Response: My apologies, the word “translation” has now been removed. The reference 143 to 142 has been corrected and all other references have been checked to correspond to the numbering in the text and tables.

This manuscript is a resubmission of an earlier submission. The following is a list of the peer review reports and author responses from that submission.

Round 1

Reviewer 1 Report

The manuscript (vaccines-1314153) is a review article, which is introducing the current preventive and therapeutic vaccine development using self-replicating (amplifying) RNA/replicon RNA and recombinant (and pseudotype) RNA virus vectors. Due to the ongoing global pandemic of coronavirus disease 2019 (COVID-19), many research groups are currently involved in the development of COVID-19 vaccines. At the same time, an increasing number of review articles regarding strategies/targets for vaccine development has been published by many researchers, including the author. In this article, the author focused on several non-segmented single-stranded RNA viruses, which have allowed us to generate full-length or subgenomic replicon systems to deliver gene fragments of foreign pathogens. In addition, the author describes regarding preventive and therapeutic vaccines for both of infectious diseases and cancers. The article may help the people, who are not so familiar with the virology but interested in new vaccine developments, roughly understand "preventive and therapeutic vaccines for infectious diseases and cancer", which are developed using several gene-modified RNA viruses. However, unfortunately, the quality of article is not enough to publish in Vaccines. The most part of the article is not a review but a list of summarized data from selected previous research. There are many errors regarding the basic virology, immunology, infectious diseases and molecular biology. In addition, there are logical fallacies and unsupported claims. Tables are not organized. There are inappropriate citations including self-citations. Further, the author only mentioned the advantages of replicon RNA vectors, which is not appropriate. English is readable but slightly informal and sometimes colloquial, and thus the article is inconsiderate to the readers, in particular non-English speakers.

Thus, unfortunately, this manuscript is not sufficient for Vaccines level.

There are too many points, which should be addressed. Some of them are listed below.

  • Abstract should include only the main point of the review. It is slightly hard to preview the article and speculate the author's point of view.
  • Introduction: It would be better to indicate the aim of the review more clearly.
  • The definition of "Self-replicating RNA virus" is not clear at all. According to the author, self-replicating RNA virus is the virus, (1) whose replicase complex, including RNA-dependent RNA polymerase (RdRp), should be "nonstructural protein" (page 1 lanes 9&10) and (2) whose genome should be replicated in the cytoplasm (page 1 lane 10, page 2 lanes 46-51). If so, measles virus (MV), vesicular stomatitis virus (VSV) and rabies virus (RV) are not self-replicating RNA viruses as their RdRps are "structural protein". In addition, more than half of the RNA viruses replicate their genome in the cytoplasm. Further, most of the publication regarding self-replicating RNA define self-replicating RNA is genome of positive-stranded RNA viruses as RNA replicon systems for negative-stranded RNA viruses usually requires additional helper virus proteins, which should be provided in trans. Please amend the manuscript.   
  • Page 1 lane 10, page 2 lane 50, "200,000-fold amplification": There is no evidence (no description) indicating that virus RNA of alphavirus is amplified approximately 2.0×105-fold in the cytosol of an infected host cell. In reference #6, the reviewer found the description and a Fig indicating "Sindbis virus particles are released from chicken cells at a rate of about 2,000 PFU/cell/h at 6-16 h post-infection." and "negative-strand RNA is synthesized only early time point of infection.". In fact, the reviewer could not find any articles clearly showing how many copies of viral RNA genome are synthesized in the infected cells. In addition, it is impossible that genome of positive-stranded RNA virus is replicated at the same rate as negative-stranded RNA virus genome. Please amend the text.
  • Page 2 57-60, "eliminating a theoretical possibility of virus genome integration into host genome": Please logically describe why the author believes that degradation of foreign RNA delivered from replicon RNA/recombinant RNA virus vectors at 5-7 days post-injection could eliminate even a theoretical possibility of its integration into host genome. FYI: SARS-CoV-2 RNA could be reverse-transcribed and integrated into the HEK293T genome using retrotransposon within 3 days post-infection (PNAS 2021).
  • Figure 1: The replicon systems are completely different between alphaviruses and other described viruses. Fig. 1 only shows alphaviruses replicon-based gene delivery system. Therefore, the caption of the Fig. 1 should be changed. In addition, electroporation is one of the transfection techniques and lipid-based transfection reagents are available for RNA replicons these days. Please amend Fig. 1.
  • Page 3 lanes 74-78 "The difference....rhabdoviruses [11].": It does not make sense. There are many reasons to generate reverse genetics system. In addition, positive-sense RNA (mRNA) could be synthesized from viral negative-sense RNA, which is not necessary to be full genome, by providing ribonucleoproteins (N, P, L proteins for MV, VSV and RV) in trans. Please logically describe why recombinant viruses are required for the efficient gene expression system based on negative-stranded RNA virus.
  • Page 3 lanes 84 & 85: Recombinant VSV was generated in 1996, not 2013 (ref#20) and VSV minigenome expression system was reported in 1995. Please amend the text.
  • References #1, #5 and #21, which are all review articles, are likely inappropriate. The sentences in lanes 30-33 and 87-90 are a kind of common knowledge. The examples of vaccine developments using RNA virus based-gene delivery systems shown in lane 37-40 are originally reported in the different research articles, not ref#5.
  • Page 3 lanes 102-104, Table 1: Virus-like particles reported in ref #29-31 are completely different from replicon particles reported in ref #26. Please read references carefully and amend the article.
  • Table 1: In the first column, the disease names should be indicated. However, the disease names are chaotically mixed with pathogen names. In addition, in the third column, the vector names should be constantly indicated. Similarly, antigen names in the second column should be constantly indicated. Further, there are full of minor errors. For examples, GPC means glycoprotein complex, not glycoprotein. PA means protective antigen, not protein antigen. Please read all references carefully and reorganized the table.
  • Section 4 Table 2: The author is likely to misunderstand the basic strategies of therapeutic vaccines for cancer. Preventive vaccines for cancer are similar to those for infectious diseases. On the other hand, therapeutic vaccines for cancer are mainly based on oncolytic virotherapy. The author should first describe what is "oncolysis". In addition, EGFP, GFP, LacZ are reporter genes, not antigen. Similarly, IFNβ, IL-12, IL-18, SLAMbind, GM-CSF.......are not antigen. Please read all references carefully and amend the manuscript.
  • Page 3 Table 1 and elsewhere, "VEE""WEE""EEE": Abbreviations, "VEE", "WEE" and "EEE" usually indicate disease names, "Venezuelan equine encephalitis", "western equine encephalitis" and "eastern equine encephalitis", not virus name. "VEE", "WEE" and "EEE" must be "VEEV", "WEEV" and "EEEV".
  • Page 1 lane 40, page 3 lanes 87, 92 and 122, page 4 lane 144 and elsewhere "target of vaccine": To the reviewer's knowledge, in the vaccine development, the term "target" usually means a part of pathogen structure or replication cycle.
  • There are many errors other than above points. Please read all references carefully, go through a whole manuscript and rewrite a whole manuscript.

Reviewer 2 Report

This review is a listing of all kinds of data sets that were obtained in vaccination approaches with - what the author calls - 'self-replicating viruses' expressing heterologous antigens.

Although the topic is important and highly interesting, the author's text is merely a dry and dense list of observed data.  The term 'self-replicating' is described as generally used - but it would be very important to include here a critical consideration of what is actually meant by this - also in distinction to 'non-self replicating viruses' (what would this be?).

Unfortunately, the data summarized by the author lack any kind of interpretation, why e.g. some vaccination experiments were successful and others not. Moreover, in some places it is stated that good vaccines exist, but it is still necessary to develop further vaccines. Why this is necessary is not explained.

Figure 1 is completely unacceptable: a reader who is not from the field is not able to understand what is depicted in this figure or what is described in the completely inadequate figure legend. The same applies to the incomplete figure legends of the tables.

The author tries to give as complete a list as possible of viruses that have been used for the purpose of vaccination against infectious diseases and tumors. However, a number of other vectors that have provided important insights are missing, such as CSFV and BVDV.

It is immensely important to describe the different requirements for tumor vaccines and infectious disease vaccines in this review.

It is a MUST to describe why about 40 years of research in this field have so far led to approval of a vaccine in only one case, against Ebola.

In its current form, the text cannot be recommended - the readership of Viruses is interested in more than just a list of empirical data.

Reviewer 3 Report

The work on said topic is in very juvenile stage as far the human trials are concerned.